# GIT-Net: Generalized Integral Transform for Operator Learning

**Chao Wang**                                                                    *chaow@nus.edu.sg*
*Department of Statistics and Data Science*
*National University of Singapore*

**Alexandre Hoang Thiery**                                                       *a.h.thiery@nus.edu.sg*
*Department of Statistics and Data Science*
*National University of Singapore*

**Reviewed on OpenReview:** *https://openreview.net/forum?id=OWKTmrVkd2*

## Abstract

This article introduces GIT-Net, a deep neural network architecture for approximating Partial Differential Equation (PDE) operators, inspired by integral transform operators. GIT-NET harnesses the fact that common differential operators commonly used for defining PDEs can often be represented parsimoniously when expressed in specialized functional bases (e.g., Fourier basis). Unlike rigid integral transforms, GIT-Net parametrizes adaptive generalized integral transforms with deep neural networks. When compared to several recently proposed alternatives, GIT-Net's computational and memory requirements scale gracefully with mesh discretizations, facilitating its application to PDE problems on complex geometries. Numerical experiments demonstrate that GIT-Net is a competitive neural network operator, exhibiting small test errors and low evaluations across a range of PDE problems. This stands in contrast to existing neural network operators, which typically excel in just one of these areas.

## 1 Introduction

Partial differential equations (PDEs) are a vital mathematical tool in the study of physical systems, providing a means to model, control, and predict the behavior of these systems. The resolution of PDEs is an essential component of many computational tasks, such as optimization of complex systems (Tröltzsch, 2010), Bayesian inference (Stuart, 2010; El Moselhy & Marzouk, 2012), and uncertainty quantification (Smith, 2013). Conventional methods for solving PDEs, such as finite element methods (Bathe, 2007), finite volume methods (Eymard et al., 2000), and finite difference methods (Liszka & Orkisz, 1980), are based on the finite-dimensional approximation of infinite-dimensional function spaces by discretizing the PDE on predefined sampling points. However, the increasing demands for PDE solvers with lower computational cost, greater flexibility on complex domains and conditions, and adaptability to different PDEs have led to the development of modern scientific computing and engineering techniques.

The recent advances in neural network methodologies, particularly their ability to process large datasets and approximate complex functions, have made them promising candidates for solving PDEs Blechschmidt & Ernst (2021). The two primary strategies to utilize neural networks in approximating PDE solutions are function learning and operator learning, both of which have proven effective at addressing PDE solver-related challenges.

**Neural networks for function learning**   Neural networks have been widely used as a powerful tool for approximating complex functions. In this framework, solutions to PDEs are parametrized by deep neural networks whose weights are obtained by minimizing an appropriate loss function (E & Yu, 2018; Raissi et al., 2019; Bar & Sochen, 2019; Smith et al., 2020; Pan & Duraisamy, 2020; Sirignano & Spiliopoulos, 2018; Zang et al., 2020; Karniadakis et al., 2021). For instance, physics-informed neural networks (PINNs) (Raissi et al., 2019) approximate solutions to PDEs by minimizing a loss functions consisting of residual terms approximating the physical model and errors for initial/boundary conditions. The theoretical understanding of this emerging class of methods is an active area of research  (He et al., 2020; E et al., 2019; Shin et al., 2020; Daubechies et al., 2022). Compared to traditional methods (eg. finite volume, finite elements), PINNs are mesh-free and can therefore be applied flexibly to a broad spectrum of PDEs without specifying a fixed space-discretization. Additionally, by exploiting high-level automatic differentiation frameworks (Bradbury et al., 2018; Paszke et al., 2017), derivative calculations and optimizations can be easily performed without considering the specific regularity properties of the PDEs. However, PINNs need to be retrained for new instances, making them inefficient in scenarios where PDEs need to be solved repeatedly, as is common for instance in (Bayesian) inverse problems (Stuart, 2010).

**Neural networks for operator learning**   Given a PDE, the operator that maps the set of input functions (eg. boundary conditions, source terms, diffusion coefficients) to the PDE solution can be approximated using deep neural networks. In prior work (Guo et al., 2016; Zhu & Zabaras, 2018; Adler & Öktem, 2017; Bhatnagar et al., 2019; Khoo et al., 2021), surrogate operators parametrized by deep convolutional neural networks (CNNs) have been employed as a mapping between finite-dimensional function spaces defined on predefined coordinate points. These methods address the need for repeated evaluations of PDEs in applications such as inverse problems, Bayesian inference, and Uncertainty Quantification (UQ). However, the training of these networks is not straightforward for meshes of varying resolutions, and for complex geometries. Furthermore, interpolation techniques are required for evaluating the solutions at coordinates outside the predefined set of coordinate points the approximate operator has been trained on. More recently, and as followed in the present article, the problem of approximating a PDE operator has been formulated as a regression problem between infinite-dimensional function spaces  (Long et al., 2018; 2019; Hesthaven & Ubbiali, 2018; Bhattacharya et al., 2021; Li et al., 2021; Lu et al., 2021; Ong et al., 2022; Gupta et al., 2021; Cao, 2021; Kissas et al., 2022; Brandstetter et al., 2022; Huang et al., 2022). The PDE-Net methodology of  Long et al. (2018; 2019) uncovers the underlying hidden PDE models by training feed-forward deep neural networks with observed dynamic data. Discretization invariant learning is studied in Ong et al. (2022); Gupta et al. (2021). Modern neural architecture designs (e.g. transformers, attention mechanisms) are applied to operator learning  (Cao, 2021; Kissas et al., 2022; Li et al., 2023). The recently proposed method MAD (Huang et al., 2022) also achieves mesh-independent evaluation through the use of a two-stage process involving training the model and finding a latent space for new instances. In the Fourier Neural Operator (FNO) of  Li et al. (2021), the Fourier transform is leveraged to efficiently parametrized integral transforms. The universal approximation property of this approach is discussed in Kovachki et al. (2021). The DeepONet approach of  Lu et al. (2021) encodes the input function space and the domain of the output function by using two separate networks; this method was later improved upon in  Lu et al. (2022) by using Principal Component Analysis (PCA), a technique also referred to as the Proper Orthogonal Decomposition (POD) or Karhunen-Loéve expansions in other communities. More recently, Bhattacharya et al. (2021) developed a general model-reduction framework consisting of defining approximate mappings between PCA-based approximations defined on the input and output function spaces. The PCA-Net neural architecture  Bhattacharya et al. (2021); Hesthaven & Ubbiali (2018) belongs to this class of methods.  An important advantage of this class of methods is that the resulting approximate operators are mesh-independent, enabling their easy application to varied settings when functions are approximated at disparate resolutions. Some of these approaches have been compared and discussed in terms of their accuracy, memory cost, and evaluation in previous work such as Lu et al. (2022); de Hoop et al. (2022). These studies indicate that the FNO approach is competitive in terms of accuracy for most PDE problems defined on structured grids, although modifications may be necessary for more complex geometries. These comparative studies also hint that DeepONet-type methods are more

flexible and accurate when used to tackle complex geometries. Finally, the PCA-NetBhattacharya et al. (2021); Hesthaven & Ubbiali (2018) architecture is more advantageous in terms of its lower memory and computational requirements. Nevertheless, as our numerical experiments presented in Section 5 indicate, the simplistic PCA-Net approach can struggle to reach high predictive accuracies when approximating complex PDE operators; we postulate that it is because the PCA-Net architecture is very general and does not incorporate any model-structure contrarily to, for instance, the FNO approach.

**Our contribution** We propose a novel neural network operator, the Generalized Integral Transform Neural Network (GIT-Net), for operator learning between infinite-dimensional function spaces serving as a solver for partial differential equations (PDEs).

- The GIT-Net architecture is derived from the remark that a large class of operators commonly used to define PDEs (eg. differential operators such as the gradient or the divergence operator) can be diagonalized in suitable functional bases (eg. Fourier basis). Compared to other widely used integral transforms such as the Fourier transform or the Laplace transform, GIT-Net takes advantage of the ability of neural networks to efficiently learn appropriate bases in a data-driven manner. This allows GIT-Net to provide improved accuracy and flexibility.

- Inspired by the model reduction techniques presented in Bhattacharya et al. (2021), GIT-Net crucially depends on Principal Component Analysis (PCA) bases of functions. As both input and output functions are represented using their PCA coefficients, the proposed method is robust to mesh discretization and enjoys favorable computational costs. When using $\mathcal{O}(N_p)$ sampling points to approximate functions, the GIT-Net approach results in an evaluation cost that scales as $\mathcal{O}(N_p)$ floating point operations, markedly more efficient than the $\mathcal{O}(N_p \log(N_p))$ evaluation costs of the FNO architecture (Li et al., 2021).

- The GIT-Net architecture is compared to modern methods for data-driven operator learning, including PCA-Net, POD-DeepONet, and FNO. For this purpose, the different methods are evaluated on five PDE problems with complex domains and input-output functions of varying dimensions. The numerical experiments suggest that when compared to existing methods, GIT-Net consistently achieves high-accuracy predictions with low evaluation costs when used for approximating the solutions to PDEs defined on rectangular grids or more complex geometries. This advantage is particularly pronounced for large-scale problems defined on complex geometries.

The rest of the paper is structured as follows. Section 2 describes the GIT-Net architecture. Section 3 provides an overview of the PDE problems used for evaluating the different methods. The hyperparameters of the GIT-Net are studied numerically and discussed in Section 4. A comparison of the GIT-Net with various other neural network operators is presented in Section 5. Finally, we present our conclusions in Section 6. Codes and datasets are publicly available [1].

## 2 Method

Consider two Hilbert spaces of functions $\mathcal{U} = \{\boldsymbol{f} : \Omega_u \to \mathbb{R}^{d_{\text{in}}}\}$ and $\mathcal{V} = \{\boldsymbol{g} : \Omega_v \to \mathbb{R}^{d_{\text{out}}}\}$ respectively defined on the input domain $\Omega_u$ and output domain $\Omega_v$. For $\boldsymbol{f} \in \mathcal{U}$ we have that $\boldsymbol{f}(\boldsymbol{x}) = (f^1(\boldsymbol{x}), \dots, f^{d_{\text{in}}}(\boldsymbol{x})) \in \mathbb{R}^{d_{\text{in}}}$ for $\boldsymbol{x} \in \Omega_u$, and similarly for functions $\boldsymbol{g} \in \mathcal{V}$. Consider a finite training set of $N_{\text{train}} \geq 1$ pairs $(\boldsymbol{f}_i, \boldsymbol{g}_i)_{i=1}^{N_{\text{train}}}$ with $\boldsymbol{g}_i = \mathcal{F}(\boldsymbol{f}_i)$ where $\mathcal{F} : \mathcal{U} \to \mathcal{V}$ denotes an unkown operator. We seek to approximate the operator $\mathcal{F}$ with the member $\mathsf{N} \equiv \mathsf{N}_\theta$ of a parametric family of operators indexed by the (finite-dimensional) parameter $\theta \in \Theta$. For a probability distribution $\mu(d\boldsymbol{f})$ of interest on the input space of functions $\mathcal{U}$, the ideal parameter $\theta_\star \in \Theta$ is chosen so that it minimizes the risk defined as

$$\theta \mapsto \mathbb{E}_{\boldsymbol{f} \sim \mu}\big[\|\mathcal{F}(\boldsymbol{f}) - \mathsf{N}_\theta(\boldsymbol{f})\|_{\mathcal{V}}^2\big]. \tag{1}$$

---

[1]Github: https://github.com/chaow-mat/General_Integral_Transform_Neural_Network

For Bayesian Inverse Problems, the distribution $\mu(d\boldsymbol{f})$ is typically chosen as the Bayesian prior distribution, or as an approximation of the Bayesian posterior distribution obtained from computationally cheap methods (eg. Gaussian-like approximations). In practice, the risk (1) is approximated by its empirical version so that the following loss function is to be minimized,

$$\mathcal{L}(\theta) = \frac{1}{N_{\text{train}}} \sum_{i=1}^{N_{\text{train}}} \|\mathcal{F}(\boldsymbol{f}_i) - \mathsf{N}_\theta(\boldsymbol{f}_i)\|_{\mathcal{V}}^2. \tag{2}$$

The loss function (2) describes a standard regression problem, although expressed in an infinite dimensional space of functions, and can be minimized with standard (variant of) the stochastic gradient descent algorithm. In the numerical experiments presented in Section 5, we used the ADAM optimizer (Kingma & Ba, 2015).

A standard method for approaching this infinite dimensional regression problem consists of expressing all the quantities on functional bases and considering finite-dimensional approximations. Let $\{\boldsymbol{e}_{u,k}\}_{k\geq 1}$ and $\{\boldsymbol{e}_{v,k}\}_{k\geq 1}$ be two bases of functions of the Hilbert spaces $\mathcal{U}$ and $\mathcal{V}$ respectively so that functions $\boldsymbol{f} \in \mathcal{U}$ and $\boldsymbol{g} \in \mathcal{V}$ can be expressed as

$$\boldsymbol{f} = \sum_{k\geq 1} \alpha_{u,k}(\boldsymbol{f})\,\boldsymbol{e}_{u,k} \qquad \text{and} \qquad \boldsymbol{g} = \sum_{k\geq 1} \alpha_{v,k}(\boldsymbol{g})\,\boldsymbol{e}_{v,k} \tag{3}$$

for coefficients $\alpha_{u,k}(\boldsymbol{f}) \in \mathbb{R}$ and $\alpha_{v,k}(\boldsymbol{g}) \in \mathbb{R}$. Truncating these expansions at a finite order provides finite-dimensional representations of each element of $\mathcal{U}$ and $\mathcal{V}$,

$$\boldsymbol{f} \;\mapsto\; [\alpha_{u,1}(\boldsymbol{f}),\ldots,\alpha_{u,N_u}(\boldsymbol{f})] \in \mathbb{R}^{N_u}$$
$$\boldsymbol{g} \;\mapsto\; [\alpha_{v,1}(\boldsymbol{g}),\ldots,\alpha_{v,N_v}(\boldsymbol{g})] \in \mathbb{R}^{N_v},$$

and transforms the infinite-dimensional regression problem (2) into a standard finite-dimensional regression setting. For example, the PCA-Net approach (Bhattacharya et al., 2021; de Hoop et al., 2022) approximates the resulting finite dimensional mapping $\widehat{\mathcal{F}} : \mathbb{R}^{N_u} \to \mathbb{R}^{N_v}$ with a standard fully-connected *multilayer perceptron* (MLP) and use Karhunen–Loéve expansions (i.e., PCA) of the probability distributions $\mu$ and its push-forward $\mathcal{F}_{\#}(\mu)$ as functional bases. Since not all Principal Components (PCs) are necessary, computing a full SVD is typically computationally wasteful. Instead, it suffices to compute the first few dominant PCs. While low-rank SVD algorithms based on the Lanczos algorithm have been available for some time, modern techniques (Halko et al., 2011) primarily rely on random projections and randomized numerical linear algebra (Martinsson & Tropp, 2020). In this text, when PCA decompositions are needed and a full SVD is too computationally costly to be implemented, we advocate utilizing these randomized techniques.

## 2.1 Generalized Integral Transform mapping

This section describes GIT-Net, a neural architecture that generalizes the Fourier Neural Operator (FNO) approach (Li et al., 2021). It also takes inspiration from the Depthwise Separable Convolutions architecture (Chollet, 2017) that has proven useful for tackling computer vision problems. Depthwise Separable Convolutions factorize convolutional filters into shared channel-wise operations that are combined in a final processing step. This factorization allows one to significantly reduce the number of learnable parameters.

For clarity, we first describe the Generalized Integral Transform (GIT) mapping in function space since the finite discretization is straightforward and amounts to considering finitely truncated basis expansions. The GIT mapping relies on the remark that mappings $\mathcal{F} : \mathcal{U} \to \mathcal{V}$ that represent solutions to *partial differential equations* (PDE) are often parsimoniously represented when expressed using appropriate bases. For example, a large class of linear PDEs can be diagonalized on the Fourier basis. Naturally, spectral methods do rely

on similar principles. The GIT mapping described in Section 2.2 is defined as a composition of mappings that transform functions $\boldsymbol{f} : \Omega \to \mathbb{R}^C$, where $C \geq 1$ denotes the number of *channels* in the computer-vision terminology, to functions $\boldsymbol{g} : \Omega \to \mathbb{R}^C$. We first describe a related linear mapping $\mathcal{K}$ that is the main component for defining the GIT mapping.

1. Assume that each component of the input function $\boldsymbol{f} : \Omega \to \mathbb{R}^C$ is decomposed on a common basis of functions $\{b_k\}_{k \geq 1}$ with $b_k : \Omega \to \mathbb{R}$. In other words, each component $f^c$ for $1 \leq c \leq C$ of the function $\boldsymbol{f} = (f^1, \ldots, f^C)$ is expressed as

$$f^c(\boldsymbol{x}) = \sum_{k \geq 1} \alpha_{c,k}\, b_k(\boldsymbol{x}) \tag{4}$$

   for coefficients $\alpha_{c,k} \in \mathbb{R}$. Define matrix $\boldsymbol{\alpha} = [\alpha_{c,k}] \in \mathbb{R}^{C,K}$. Note that it is different from expanding the function $\boldsymbol{f}$ on a basis of $\mathbb{R}^C$-valued functions. In contrast, the GIT-Net approach proceeds by expanding each component $f^c : \Omega \to \mathbb{R}$ on a common basis of real-valued functions.

2. A linear change of basis is implemented. The expansion on the basis of functions $\{b_k\}_{k \geq 1}$ is transformed into an expansion onto another basis of functions $\{\check{b}_k\}_{k \geq 1}$ with $\check{b}_k : \Omega \to \mathbb{R}$. Heuristically, this can be thought of as the equivalent of expressing everything on a Fourier basis, i.e., a Fourier transform. We have

$$f^c(\boldsymbol{x}) \;=\; \sum_{k \geq 1} \check{\alpha}_{c,k}\, \check{b}_k(\boldsymbol{x}) \tag{5}$$

   for coefficients $\check{\alpha}_{c,k} \in \mathbb{R}$. Define matrix $\check{\boldsymbol{\alpha}} = [\check{\alpha}_{c,K}] \in \mathbb{R}^{C,k}$. Continuing the analogy with Fourier expansions, the coefficients $\{\check{\alpha}_{c,k}\}_{k \geq 1}$ represent the Fourier-spectrum of the function $f^c : \Omega \to \mathbb{R}$. The mapping $\boldsymbol{\alpha} \mapsto \check{\boldsymbol{\alpha}}$ is linear: it represents a standard (linear) change of basis.

3. The main assumption of the GIT-Net transform is that the different frequencies do not interact with each other. In the degenerate case $C = 1$, this corresponds to an operator that is diagonalized in the Fourier-like basis $\{\check{b}_k\}_{k \geq 1}$. The function $\boldsymbol{g} = \mathcal{K}\boldsymbol{f}$ with $\boldsymbol{g} = (g^1, \ldots, g^C)$ is defined as

$$g^c(\boldsymbol{x}) \;=\; \sum_{k \geq 1} \check{\beta}_{c,k}\, \check{b}_k(\boldsymbol{x}) \qquad \text{where} \qquad \check{\beta}_{c,k} = \sum_{d=1}^{C} \mathbf{D}_{d,c,k}\, \check{\alpha}_{d,k} \tag{6}$$

   for coefficients $\mathbf{D}_{d,c,k} \in \mathbb{R}$. The coefficients $\mathbf{D}_{d,c,k}$ for $1 \leq c, d \leq C$ and $k \geq 1$ represent the linear mapping $\check{\boldsymbol{\alpha}} \mapsto \check{\boldsymbol{\beta}}$. Crucially, for each frequency index $k \geq 1$, the frequencies $\{\check{\beta}_{c,k}\}_{c=1}^{C}$ are linear combinations of the frequencies $\{\check{\alpha}_{c,k}\}_{c=1}^{C}$ only.

4. As a final step, a last change of basis is performed. Heuristically, this can be thought of as the equivalent of implementing an inverse Fourier transform in order to come back to the original basis. More generally, the proposed approach only assumes a linear change of basis and expresses all the quantities in a final basis of functions $\{\widehat{b}_k\}_{k \geq 1}$ with $\widehat{b}_k : \Omega \to \mathbb{R}$. The basis $\{\widehat{b}_k\}_{k \geq 1}$ is not assumed to be the same as the original basis $\{b_k\}_{k \geq 1}$. We have

$$g^c(\boldsymbol{x}) \;=\; \sum_{k \geq 1} \beta_{c,k}\, \widehat{b}_k(\boldsymbol{x}) \tag{7}$$

   for coefficients $\beta_{c,k} \in \mathbb{R}$. The mapping $\check{\boldsymbol{\beta}} \mapsto \boldsymbol{\beta}$ is linear.

The operations described above, when expressed in finite bases of size $K \geq 1$ instead of infinite expansions, can be summarized as follows. The input function $\boldsymbol{f} : \Omega \to \mathbb{R}^C$ is represented as $\boldsymbol{\alpha} \equiv [\alpha_{c,k}] \in \mathbb{R}^{C,K}$ when the coordinate functions are expanded on the finite basis $\{b_k\}_{k=1}^{K}$. After the change of basis $\{b_k\}_{k=1}^{K} \mapsto \{\check{b}_k\}_{k=1}^{K}$,

the function is represented as $\check{\alpha} \equiv [\check{\alpha}_{c,k}] \in \mathbb{R}^{C,K}$. This linear change of basis can be implemented by the right-multiplication by a matrix $\mathbf{P} \in \mathbb{R}^{K,K}$ so that $\check{\alpha} = \alpha \mathbf{P}$. To succinctly describe the next operation, we use the notation $\otimes$ to denote the frequency-wise operation defined in Equation (6). For $\check{\boldsymbol{\alpha}} \in \mathbb{R}^{C,K}$ and a tensor $\mathbf{D} = [\mathbf{D}_{d,c,k}] \in \mathbb{R}^{C,C,K}$, the operation $\check{\boldsymbol{\beta}} = \check{\boldsymbol{\alpha}} \otimes \mathbf{D}$ with $\check{\boldsymbol{\beta}} \in \mathbb{R}^{C,K}$ is defined as

$$\check{\beta}_{c,k} = \sum_{d=1}^{C} \mathbf{D}_{d,c,k}\, \check{\alpha}_{d,k}$$

for any $1 \le c \le C$ and $1 \le k \le K$. Finaly, the last change of basis $\{\check{b}_k\}_{k=1}^{K} \mapsto \{\widehat{b}_k\}_{k=1}^{K}$ can be implemented by the right-multiplication by a matrix $\mathbf{Q} \in \mathbb{R}^{K,K}$. The composition of these three (linear) operations, namely $\boldsymbol{\beta} = \mathcal{K}\alpha$ with

$$\mathcal{K}\boldsymbol{\alpha} \;\equiv\; ((\boldsymbol{\alpha}\,\mathbf{P}) \otimes \mathbf{D})\,\mathbf{Q}, \tag{8}$$

is a generalization of the *non-local operator* introduced in Li et al. (2021). For clarity, we have used the same notation $\mathcal{K}$ to denote the operator mapping $\boldsymbol{g} = \mathcal{K}\boldsymbol{f}$ and the corresponding discretization $\boldsymbol{\beta} = \mathcal{K}\boldsymbol{\alpha}$ when expressed on finite bases. Note that the set of such generalized non-local operators is parametrized by a set of $(2\,K^2 + K\,C^2)$ parameters while the vector space of all linear transformations from $\mathbb{R}^{C,K}$ onto itself has dimension $(KC)^2$. The (nonlinear) GIT mapping is defined as

$$\mathsf{G}(\boldsymbol{\alpha}) \;=\; \sigma(\mathbf{T}\,\boldsymbol{\alpha} + \mathcal{K}\boldsymbol{\alpha}) \tag{9}$$

for a matrix $\mathbf{T} \in \mathbb{R}^{C,C}$ and a non-linear activation function $\sigma : \mathbb{R} \to \mathbb{R}$ applied component-wise. The numerical experiments presented in Section 5 use the GELU nonlinearity (Hendrycks & Gimpel, 2016) although other choices are certainly possible. A more intuitive diagram of the GIT mapping and corresponding tensor's dimensions are shown in Figure 1. Introducing the term $\mathbf{T}\,\boldsymbol{\alpha}$ does not make the transformation

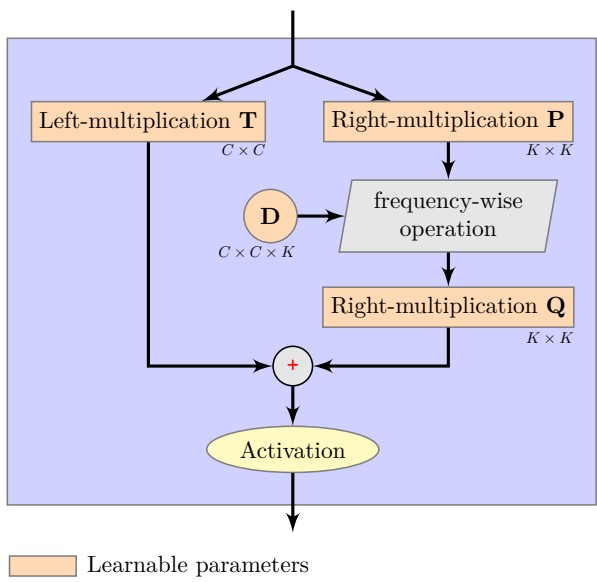

Figure 1: GIT mapping $\mathsf{G}$ and corresponding tensor's dimensions.

more expressive when compared to only using the transformation $\boldsymbol{\alpha} \mapsto \sigma(\mathcal{K}\boldsymbol{\alpha})$ since it is possible to define $\widetilde{\mathcal{K}}\boldsymbol{\alpha} \equiv ((\boldsymbol{\alpha}\,\widetilde{\mathbf{P}}) \otimes \widetilde{\mathbf{D}})\,\widetilde{\mathbf{Q}}$ for another set of parameters $(\widetilde{\mathbf{P}}, \widetilde{\mathbf{D}}, \widetilde{\mathbf{Q}})$ so that $\mathsf{G}(\boldsymbol{\alpha}) = \sigma(\widetilde{\mathcal{K}}\boldsymbol{\alpha})$. Nevertheless, we have empirically observed that the over-parametrization defined in Equation (9) eases the optimization process

and helps the model reach higher predictive accuracies. It is worth emphasizing that the GIT-Net mapping, as described in Section 2.2, does not attempt to learn the basis functions $b_k$ and $\check{b}_k$ and $\hat{b}_k$. Another similar transformation $\mathbf{T}\,\boldsymbol{\alpha} + \sigma(\mathcal{K}\boldsymbol{\alpha})$ is also explored and compared with (9) numerically for all PDE problems. It is shown that they achieve similar prediction errors in Appendix B.

## 2.2 GIT-Net: Generalized Integral Transform Neural Network

This section describes the full GIT-Net architecture whose main component is the GIT mapping described in Equation (9). Recall that we are given a training set of pairs of functions $\{(\boldsymbol{f}_i, \boldsymbol{g}_i)\}_{i=1}^{N_{\text{train}}}$ where $\boldsymbol{g}_i = \mathcal{F}(\boldsymbol{f}_i)$ for some unknown operator $\mathcal{F} : \mathcal{U} \to \mathcal{V}$ with $\boldsymbol{f}_i : \Omega_u \to \mathbb{R}^{d_{\text{in}}}$ and $\boldsymbol{g}_i : \Omega_v \to \mathbb{R}^{d_{\text{out}}}$. The purpose of the GIT-Net architecture is to approximate the unknown operator $\mathcal{F}$.

In order to transform this infinite dimensional regression problem into a finite one, we assume a set of functions $(e_{u,1}, \ldots, e_{u,P_u})$ with $e_{u,i} : \Omega_u \to \mathbb{R}$, as well as a set of functions $(e_{v,1}, \ldots, e_{v,P_v})$ with $e_{v,i} : \Omega_v \to \mathbb{R}$. This allows one to define the approximation operator $\mathcal{A}_u : \mathcal{U} \to \mathbb{R}^{d_{\text{in}}, P_u}$ such that, for a function $\boldsymbol{f} = (f^1, \ldots, f^{d_{\text{in}}})$, we have

$$f^c \approx \sum_{k=1}^{P_u} \alpha_{c,k}\, e_{u,k} \qquad \text{for} \quad 1 \le c \le d_{\text{in}}$$

and coefficents $\boldsymbol{\alpha} \in \mathbb{R}^{d_{\text{in}}, P_u}$ given by $\boldsymbol{\alpha} = \mathcal{A}_u(\boldsymbol{f})$; the approximation operator $\mathcal{A}_v : \mathcal{V} \to \mathbb{R}^{d_{\text{out}}, P_v}$ is defined similarly. For completeness, we define the operator $\mathcal{P}_u : \mathbb{R}^{d_{\text{in}}, P_u} \to \mathcal{U}$ that sends $\boldsymbol{\alpha} \in \mathbb{R}^{d_{\text{in}}, P_u}$ to the function $(\overline{f}^1, \ldots, \overline{f}^{d_{\text{in}}}) = \overline{\boldsymbol{f}} = \mathcal{P}_u(\boldsymbol{\alpha})$ defined as $\overline{f}^c = \sum_{k=1}^{N} \alpha_{c,k}\, e_{u,k}$. The operator $\mathcal{P}_v : \mathbb{R}^{d_{\text{out}}, P_v} \to \mathcal{V}$ is defined similarly so that

$$\boldsymbol{f} \approx \mathcal{P}_u \circ \mathcal{A}_u\,(\boldsymbol{f}) \qquad \text{and} \qquad \boldsymbol{g} \approx \mathcal{P}_v \circ \mathcal{A}_v\,(\boldsymbol{g})$$

for any functions $\boldsymbol{f} \in \mathcal{U}$ and $\boldsymbol{g} \in \mathcal{V}$. The mappings $\mathcal{P}_u \circ \mathcal{A}_u$ and $\mathcal{P}_v \circ \mathcal{A}_v$ can be thought of as simple linear auto-encoders (Rumelhart et al., 1985); naturally, more sophisticated (eg. non-linear) representations can certainly be used although we have not explored that direction further. In all the applications presented in this text, we have used PCA orthonormal bases obtained from the finite training set of functions; the approximation operators $\mathcal{A}_u$ and $\mathcal{A}_v$ are the orthogonal projections of these PCA bases. Numerically, given discretized input functions $\{\boldsymbol{F}_i\}_{i=1}^{N_{\text{train}}}$ with $\boldsymbol{F} \in \mathbb{R}^{N \times d_{in}}$, PCA bases are computed from matrix $[\boldsymbol{F}_1, \boldsymbol{F}_2, \ldots, \boldsymbol{F}_{N_{\text{train}}}]$. Similarly, PCA bases of discretized output functions can be computed. Other standard finite dimensional approximations (eg. finite-element, dictionary learning, spectral methods) could have been used instead. Through these approximation operators, the infinite-dimensional regression problem is transformed into a finite-dimensional problem consisting of predicting $\mathcal{A}_v[\mathcal{F}(\boldsymbol{f})] \in \mathbb{R}^{d_{\text{out}}, P_v}$ from the input $\mathcal{A}_v[\boldsymbol{f}] \in \mathbb{R}^{d_{\text{in}}, P_u}$.

To leverage the GIT mapping (9), we consider a lifting operator $\Phi^{\uparrow} : \mathbb{R}^{d_{\text{in}}, P_u} \to \mathbb{R}^{C,K}$ that maps $\boldsymbol{\alpha} \in \mathbb{R}^{d_{\text{in}}, P_u}$ to $\Phi^{\uparrow}(\boldsymbol{\alpha}) \in \mathbb{R}^{C,K}$ with a number of channel $C \ge 1$ possibly significantly higher than $d_{\text{in}}$. Similarly, we consider a projection operator $\Phi^{\downarrow} : \mathbb{R}^{C,K} \to \mathbb{R}^{d_{\text{out}}, P_v}$. In the numerical experiments presented in Sections 4 and 5, the lifting and projection operators were chosen as simple linear operators defined as

$$\Phi^{\uparrow}(\boldsymbol{\alpha}) = \mathbf{L}^{\uparrow}\,\boldsymbol{\alpha}\,\mathbf{R}^{\uparrow} \qquad \text{and} \qquad \Phi^{\downarrow}(\boldsymbol{\alpha}) = \mathbf{L}^{\downarrow}\,\boldsymbol{\alpha}\,\mathbf{R}^{\downarrow}$$

for (learnable) matrices $\mathbf{L}^{\uparrow} \in \mathbb{R}^{C, d_{\text{in}}}$ and $\mathbf{R}^{\uparrow} \in \mathbb{R}^{P_u, K}$ and $\mathbf{L}^{\downarrow} \in \mathbb{R}^{d_{\text{out}}, K}$ and $\mathbf{R}^{\downarrow} \in \mathbb{R}^{K, P_v}$; we chose $P_u = P_v = K$, but that is not a requirement. The unknown operator $\mathcal{F} : \mathcal{U} \to \mathcal{V}$ is approximated by the GIT-Net neural network $\mathsf{N} : \mathcal{U} \to \mathcal{V}$ depicted in Figure 2 and defined as

$$\mathsf{N} \equiv \mathcal{P}_v \circ \Phi^{\downarrow} \circ \underbrace{\mathsf{G}^{(L)} \circ \ldots \circ \mathsf{G}^{(1)}}_{L \text{ transformations}} \circ \Phi^{\uparrow} \circ \mathcal{A}_u. \tag{10}$$

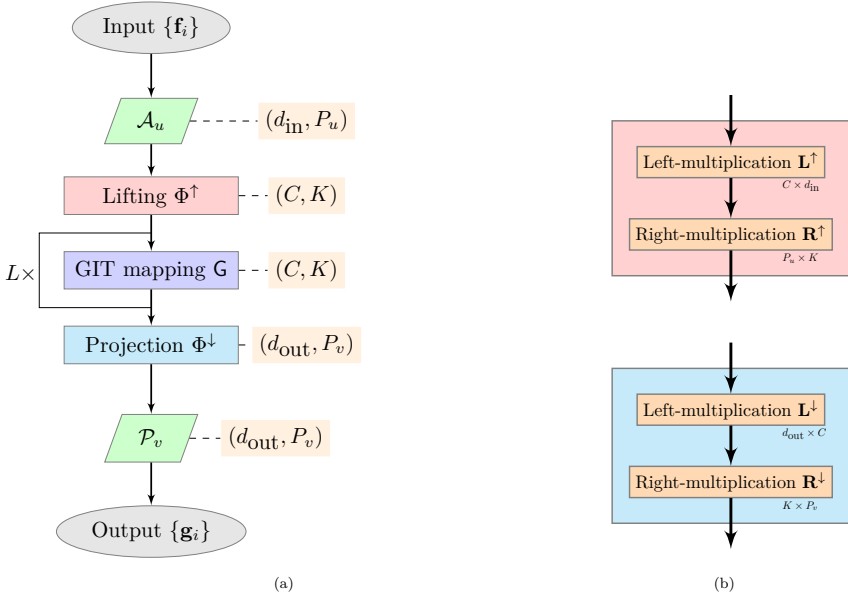

Figure 2: (a) The architecture of an L-layer GIT-Net. The dimension of data after each operation is shown. (b) Top: lifting layer; bottom: projection layer. The blocks in orange are learnable parameters.

The GIT-Net operator $\mathsf{N} : \mathcal{U} \to \mathcal{V}$ is defined by composing $L \geq 1$ different GIT-Net layers (9); nonlinear activation functions $\sigma$ are used for $\mathsf{G}^{(l)}(l = 1, \ldots, L-1)$ and identity activation function is used instead in the last layer $\mathsf{G}^{(L)}$. The operator first approximates each component of the input function $\boldsymbol{f} = (f^1, \ldots, f^{d_{\mathrm{in}}})$ as a linear combination of $P_u \geq 1$ functions $(e_{u,1}, \ldots, e_{u,P_u})$ thanks to the approximation operator $\mathcal{A}_u$. The finite representation $\mathcal{A}_u(\boldsymbol{f}) \in \mathbb{R}^{d_{\mathrm{in}},P_u}$ is then lifted to a higher dimensional space of dimension $\mathbb{R}^{C,K}$ thanks to the lifting operator $\Phi^\uparrow$, before going through $L \geq 1$ nonlinear GIT mappings (9). Finally, the representation is projected down to $\mathbb{R}^{d_{\mathrm{out}},P_v}$ in order to reconstruct the output $\boldsymbol{g} = \mathsf{N}(\boldsymbol{f}) \in \mathcal{V}$. Each coordinate of the output function $\boldsymbol{g}$ is expressed as a linear combination of the functions $(e_{v,1}, \ldots, e_{v,P_v})$ through the operator $\mathcal{P}_v$. As explained in Section 2, the learnable parameters are obtained by minimizing the empirical risk defined in Equation (1).

## 3   PDE problems

In this section, we introduce the PDE problems employed to evaluate the performance of GIT-Net in comparison with other established methods. These PDEs are defined on domains ranging from simple rectangular grids to more complex geometries. The numerical experiments include the Navier-Stokes equation, the Helmholtz equation, the advection equation, along with the Poisson equation. The datasets for the Navier-Stokes, Helmholtz, and advection equations are identical to those used in the work of de Hoop et al. (2022). The dataset for the Poisson equation follows the setup described in the work of Lu et al. (2022).

**Navier-Stokes equation**   Following the setup of de Hoop et al. (2022), the incompressible Navier-Strokes equation is expressed in the vorticity-stream form on the two-dimensional periodic domain $D = [0, 2\pi]^2$. It

reads as

$$
\begin{cases}
\left(\dfrac{\partial \omega}{\partial t} + (v \cdot \nabla)\omega - \nu\Delta\omega\right)(\boldsymbol{x},t) = f(\boldsymbol{x}), & (\boldsymbol{x},t) \in D \times [0,T], \\[2mm]
\omega = -\Delta\psi, \quad \displaystyle\int_D \psi = 0, & (\boldsymbol{x},t) \in D \times [0,T], \\[2mm]
v = \left(\dfrac{\partial\psi}{\partial x_2}, -\dfrac{\partial\psi}{\partial x_1}\right), & (\boldsymbol{x},t) \in D \times [0,T], \\[2mm]
\omega(\boldsymbol{x},0) = \omega_0(\boldsymbol{x}), & \boldsymbol{x} \in D,
\end{cases}
\tag{11}
$$

where the quantities $f(\boldsymbol{x})$ and $\omega(\boldsymbol{x},t)$ and $v(\boldsymbol{x},t)$ represent the forcing, the vorticity field, and the velocity field, respectively. The viscosity $\nu$ is fixed at 0.025 and the final time is set to $T = 10$. In this study, we investigate the mapping $f(\boldsymbol{x}) \mapsto \omega(\boldsymbol{x},T)$ for $\boldsymbol{x} \in D$. The input function $f$ is a sample from a Gaussian random field $\mathcal{N}(0,\Gamma)$, where the covariance operator is given by $\Gamma = (-\Delta + 9)^{-2}$ and $\Delta$ is the Laplacian operator on the domain $D = [0,2\pi]^2$ with periodic boundary conditions. The output function $\omega(\cdot,T)$ is obtained by solving Equation (11) with the initial condition $\omega_0(\boldsymbol{x})$. The initial condition $\omega_0$ is generated from the same distribution as $f$. For solving this equation, the domain is discretized on a uniform $64 \times 64$. Further details can be found in de Hoop et al. (2022). The same training and testing sets as de Hoop et al. (2022) were used in the numerical experiments. Figure 3 illustrates a pair of input and output functions.

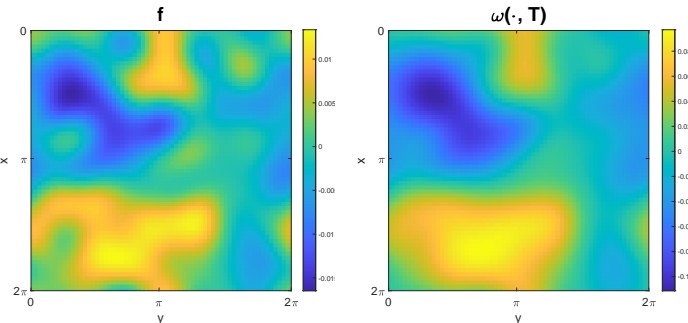

Figure 3: Navier-Stokes Equation (11). **Left:** input force $f(\boldsymbol{x})$. **Right:** final vorticity field $\omega(\boldsymbol{x},T)$.

**Helmholtz equation** Following the setup of de Hoop et al. (2022), Equation (12) describes the Helmholtz equation on the two-dimensional domain $D = [0,1]^2$,

$$
\begin{cases}
\left(-\Delta - \dfrac{\omega^2}{c^2(\boldsymbol{x})}\right)u = 0, & \boldsymbol{x} \in D, \\[2mm]
\dfrac{\partial u}{\partial n}(\boldsymbol{x}) = u_N(\boldsymbol{x}), & \boldsymbol{x} \in \mathcal{B} \\[2mm]
\dfrac{\partial u}{\partial n}(\boldsymbol{x}) = 0, & \boldsymbol{x} \in \partial D \setminus \mathcal{B}
\end{cases}
\tag{12}
$$

where $\mathcal{B} = \{(x,1) \; : \; x \in [0,1]\} \subset \partial D$. The quantity $c(\boldsymbol{x})$ is the wavespeed field and the frequency is set to $\omega = 10^3$. The boundary condition is given by $u_N$ is $1_{\{0.35 \leq x_1 \leq 0.65\}}$. We are interested in approximating the mapping $c(\boldsymbol{x}) \mapsto u(\boldsymbol{x})$. The wavespeed is defined as $c(\boldsymbol{x}) = 20 + \tanh(\xi)$ where the quantity $\xi : D \to \mathbb{R}$ is sampled from the Gaussian random field $\xi \sim \mathcal{N}(0,\Gamma)$, where the covariance operator is given by $\Gamma = (-\Delta + 9)^{-2}$ and $\Delta$ is the Laplacian operator on the domain on $D$ with the homogeneous Neumann boundary conditions. The solution $u : D \to \mathbb{R}$ is obtained by solving (12) using the finite element method on a uniform grid of size $101 \times 101$. Numerical experiments were implemented using a dataset identical to the one presented in de Hoop et al. (2022) for training and testing. Figure 4 shows an example of wavespeed $c : D \to \mathbb{R}$ and associated solution $u : D \to \mathbb{R}$.

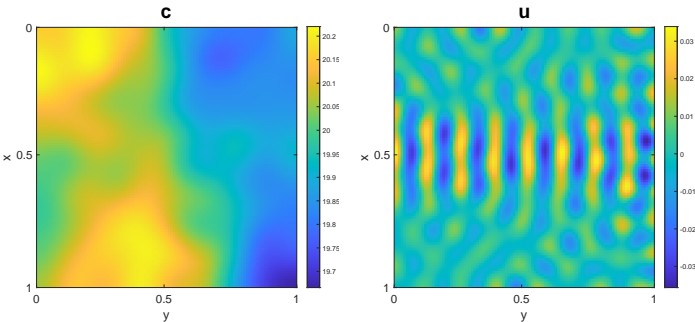

Figure 4: Helmholtz Equation (12). **Left**: (input) wavespeed field $c(\boldsymbol{x})$. **Right**: (output) solution $u(\boldsymbol{x})$.

**Structural mechanics equation**  We consider a two-dimensional plane-stress elasticity problem (Slaughter, 2012) on the rectangular domain $D = [0,1]^2 \setminus \mathcal{H}$ with a round hole $\mathcal{H}$ removed in the center, as depicted in Figure 5. It is defined as

$$
\begin{cases}
\nabla \cdot \boldsymbol{\sigma} = 0, & \boldsymbol{x} \in D, \\
\boldsymbol{\varepsilon} = \dfrac{1}{2}(\nabla \boldsymbol{u} + (\nabla \boldsymbol{u})^\top), & \boldsymbol{x} \in D, \\
\boldsymbol{\sigma} = \mathbf{C} : \boldsymbol{\varepsilon}, & \boldsymbol{x} \in D, \\
\boldsymbol{\sigma} \cdot \boldsymbol{n} = b(\boldsymbol{x}), & \boldsymbol{x} \in \mathcal{B}, \\
\boldsymbol{u}(\boldsymbol{x}) = 0, & \boldsymbol{x} \in \partial D \setminus \mathcal{B}
\end{cases}
\tag{13}
$$

for displacement vector $\boldsymbol{u}$, Cauchy stress tensor $\boldsymbol{\sigma}$, infinitesimal strain tensor $\boldsymbol{\varepsilon}$, fourth-order stiffness tensor $\mathbf{C}$ and $\mathcal{B} \equiv \{(x,1) \ : \ x \in [0,1]\} \subset \partial D$. The notation ':' denotes the inner product between two second-order tensors (summation over repeated indices is implied). The Young's modulus is set to $2 \times 10^5$ and the Poisson's ratio to 0.25. We are interested in approximating the mapping from the boundary condition to the von Mises stress field. The input boundary condition $b(x)$ is sample from the Gaussian random field $\mathcal{N}(100, \Gamma)$ with covariance operator $\Gamma = 400^2 \left(-\Delta + 9\right)^{-1})$ where $\Delta$ is the Laplacian on the domain $D$ with Neumann boundary conditions. Equation (13) is solved with the finite element method using a triangular mesh with 4072 nodes and 1944 elements. When implementing the FNO method that requires a uniform rectangular grid, the functions are interpolated to $[0,1]^2$ using a uniform grid of size $101 \times 101$. Figure 5 shows an example of input and output.

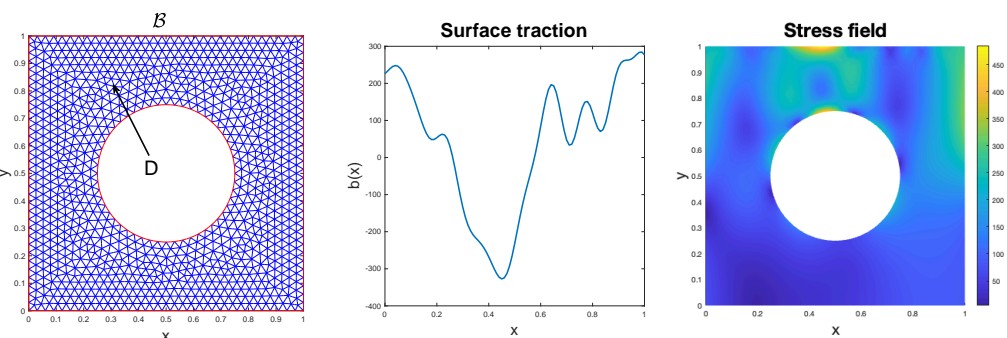

Figure 5: Structural mechanics Equation (13). Left to Right: discretization of the domain, surface traction on $\Omega$; corresponding von Mises stress field.

**Advection equation** Following the setup described in de Hoop et al. (2022), we consider the advection equation defined in the one-dimensional torus $D = [0, 1) \equiv \mathbb{R}/\mathbb{Z}$,

$$\begin{cases} \dfrac{\partial u}{\partial t} + c\dfrac{\partial u}{\partial x} = 0, & (x, t) \in D \times (0, T], \\ u(0, t) = u(1, t), & t \in [0, T] \\ u(x, 0) = u_0(x), & x \in D \end{cases} \tag{14}$$

with constant advection speed $c = 1$ and periodic boundary conditions. The initial condition is set to $u_0 = \text{sign}(\xi)$ where $\text{sign} : \mathbb{R} \to \{-1, 1\} \subset \mathbb{R}$ is the sign function and $\xi : D \to \mathbb{R}$ is sampled from the Gaussian random field $\xi \sim \mathcal{N}(0, \Gamma)$ with covariance operator $\Gamma \equiv (-\Delta + 9)^{-2}$ where $\Delta$ is the Laplacian on the domain $D$ with periodic boundaries. We are interested in approximating the mapping from initial condition $u_0(x)$ to solution $u(x, T)$ at time $T = 0.5$. We use a dataset identical to the one used in de Hoop et al. (2022) for training and testing. Figure 6 shows an example of input condition $u_0 : D \to \mathbb{R}$ and output solution $u(x, T)$.

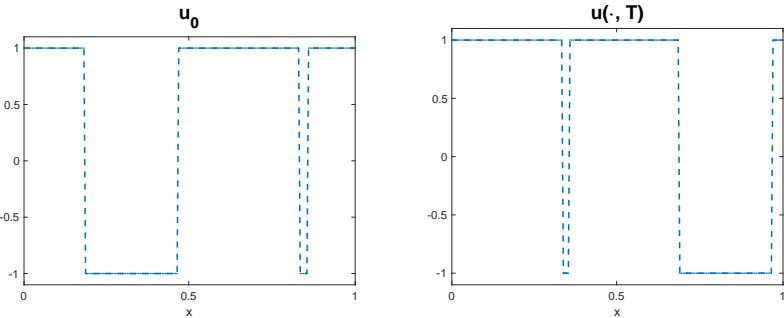

Figure 6: Advection Equation (14). **Left**: initial condition $u_0(\boldsymbol{x})$ **Right**: solution $u(\boldsymbol{x}, T)$.

**Poisson equation** Consider the two-dimensional Poisson equation defined on a triangular domain $D \subset [0, 1]^2$ with a notch, as described in Lu et al. (2022) and depicted in Figure 7. It reads

$$-\Delta h(\boldsymbol{x}) = f, \qquad \boldsymbol{x} \in D, \tag{15}$$

with pressure field $h : D \to \mathbb{R}$ and constant source term $f = -1$. The region of the notch is denoted by $\Omega$ and the condition $h(\boldsymbol{x})|_{\partial D \cap \partial \Omega} = 0$ is imposed. We are interested in approximating the mapping from the boundary condition $h(\boldsymbol{x})|_{\partial D \setminus \partial \Omega}$ to the internal pressure field $h(\boldsymbol{x})|_D$. The input boundary conditions $h(\boldsymbol{x})|_{\partial D \setminus \partial \Omega}$ need to be specified on the triangle $\partial D \setminus \partial \Omega$. As in Lu et al. (2022), on each side of this triangle, the conditions are sampled from a one-dimensional centered Gaussian Process (GP) with RBF covariance kernel $\mathcal{R}(x, y) = \exp[-(x - y)^2 / (2\ell^2)]$ with lengthscale $\ell = 0.2$. The internal pressure field $h(\boldsymbol{x})$ for $\boldsymbol{x} \in D$ is obtained by solving Equation (15) using a triangular mesh with 2295 nodes and 1082 elements, as represented in Figure 7. Since the FNO method requires a uniform rectangular grid, functions are extrapolated to $[0, 1]^2$ using a $101 \times 101$ grid and linear interpolation when outside of $D$. The reader is referred to Lu et al. (2022) for details. Figure 7 shows an example of a boundary condition and associated internal pressure field.

## 4 Influence of the hyperparameters

This section investigates the influence of the number of channels $C \geq 1$ as well as the dimension $K \geq 1$ when using the GIT-Net approach for solving PDE problems introduced in Section 3. The number $L \geq 1$ of GIT layers is fixed at $L = 3$ and we used $C \in \{2, 4, 8, 16, 32\}$ and $K \in \{16, 64, 128, 256, 512\}$. The number of PCA basis functions is set by ensuring that 99.999% of the energy is preserved. If this number

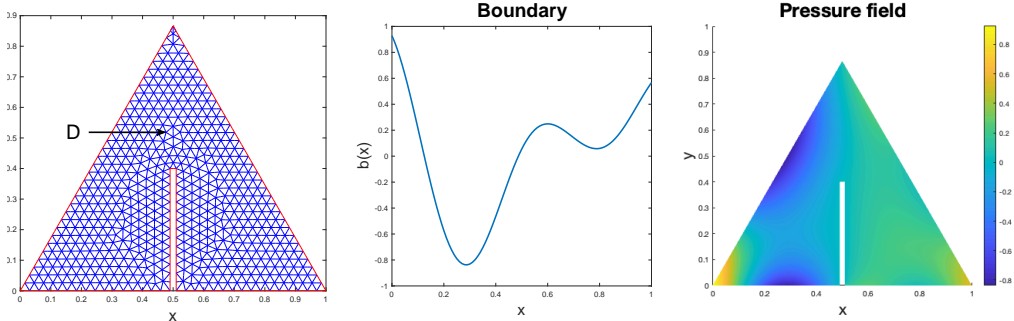

Figure 7: Poisson equation (15) in a triangular domain with a notch. Left to right are discretization of the domain $D$; boundary condition $h(\boldsymbol{x})|_{\partial D/\partial \Omega}$; corresponding internal pressure field $h(\boldsymbol{x})$.

of PCA basis functions is larger than $P = 200$, it is truncated at $P = 200$ in order to prevent the model from becoming excessively large. For the Navier-Stokes equation, Helmholtz equation, structural mechanics equation, advection equation, and Poisson equation, the number of PCA basis functions used as input-output in the network are 200-200, 200-200, 28-200, 198-198, and 16-16, respectively. To evaluate the performance of the different methods, four training datasets of respective size $N_{\text{train}} \in \{2500, 5000, 10000, 20000\}$ were generated for training; the methods were evaluated on independent test sets. The GELU nonlinear activation function (Hendrycks & Gimpel, 2016) was used to implement the GIT-Net. Let $N = N_{\text{train}} = N_{\text{test}}$. Figure 8 reports the relative test error defined as

$$E_{\text{test}} = \frac{1}{N_{\text{test}}} \sum_{i=1}^{N_{\text{test}}} \frac{\|\mathsf{N}(\boldsymbol{f}_i) - \boldsymbol{g}_i\|_2}{\|\boldsymbol{g}_i\|_2}. \tag{16}$$

as a function of the size of the training set, as well as the number of channels $C \geq 1$ and the dimension $K \geq 1$. In the small dataset regime $N_{\text{train}} = 2500$, there is overfitting for all PDE problems, with particularly severe overfitting in the Helmholtz equation and Poisson equation. In general, a smaller value of $C$ leads to more overfitting when increasing $K$, as seen in the cases of the advection and Helmholtz equation with $N_{\text{train}} = 2500$ training data. As expected, overfitting as a function of the dimensional parameter $K \geq 1$ is decreased as the size of the training size is increased. For the Navier-Stokes problem, the dimension $K$ appears to play a more important role than the number of channels $C$. For $K = 16$ the relative test error is insensitive to the number of channels. It is only when $K \geq 64$ that increasing the number of channels $C$ helps to decrease the relative test errors. However, for the Poisson equation, the number of channels $C$ appears to play a more important role. The largest value of $C$, i.e., $C = 32$, corresponds to the smallest relative test error even for the smallest value of $K$, i.e., $K = 16$.

## 5   Comparison with existing neural network operators

This section compares the GIT-Net architecture to three recently proposed baseline approaches: the PCA-Net architecture (Li et al., 2021; Bhattacharya et al., 2021; de Hoop et al., 2022), the POD-DeepONet architecture (Lu et al., 2022) and the Fourier neural operator (FNO) approach (Li et al., 2021). These neural architectures are briefly summarized in Section 5.1 for completeness. Various hyperparameters were tested for each one of these methods and Section 5.2 reports the predictions corresponding to the hyperparameters with the smallest relative test errors. Additionally, in order to demonstrate the computational efficiency of the GIT-Net architecture, Section 5.2 compares the evaluation cost of these operators in terms of floating point operations.

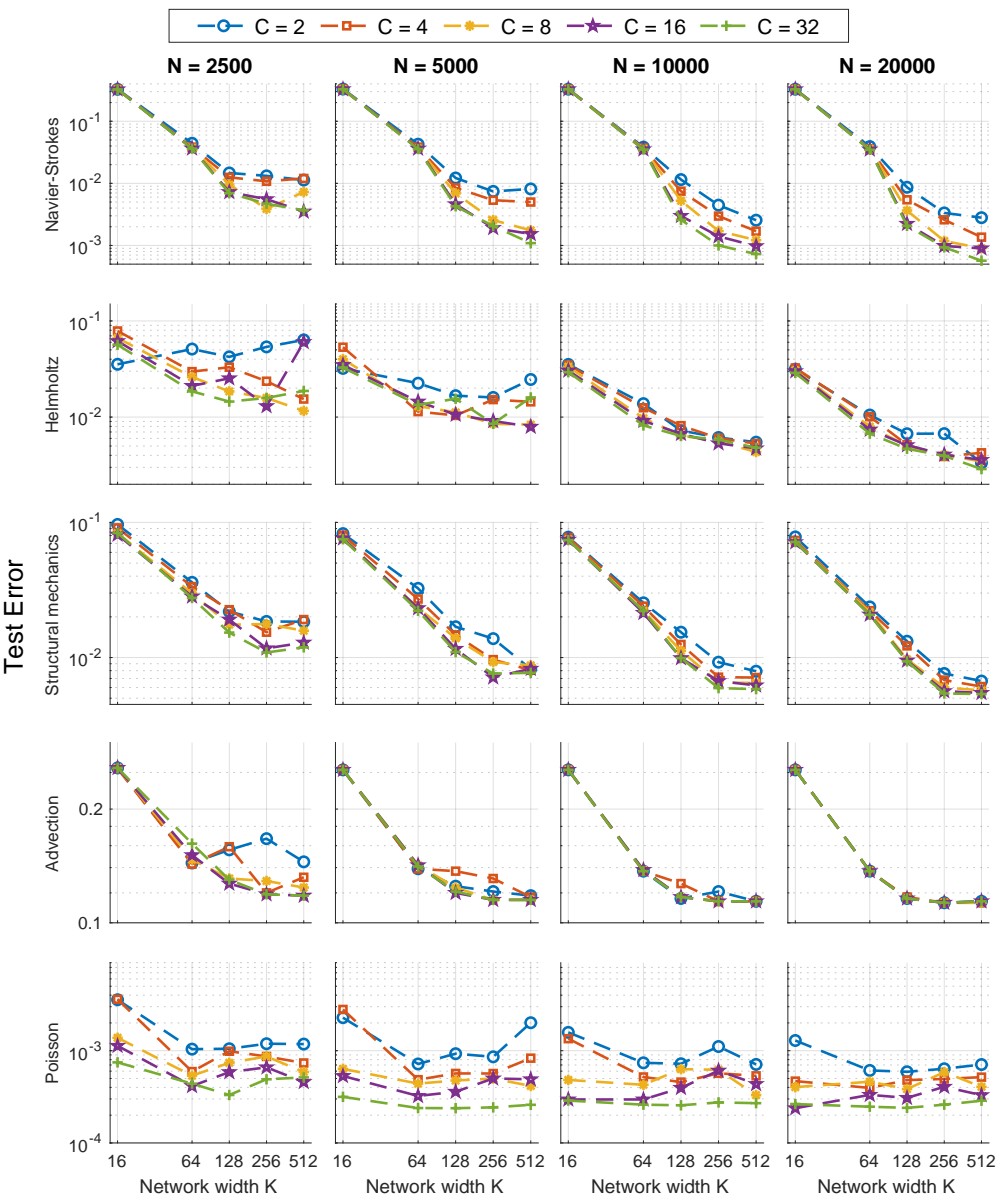

Figure 8: Relative test error as defined in Equation (16).

## 5.1 Neural architectures for operator learning

This section briefly summarizes the PCA-Net architecture, the POD-DeepONet architecture as well as the Fourier neural operator (FNO) approach. These neural architectures have recently been proposed for approximating infinite-dimensional operators. As described in Section 2, we consider two Hilbert spaces of

functions $\mathcal{U} = \boldsymbol{f} : \Omega_u \to \mathbb{R}^{d_{\text{in}}}$ and $\mathcal{V} = \boldsymbol{g} : \Omega_v \to \mathbb{R}^{d_{\text{out}}}$, defined on the input domain $\Omega_u$ and output domain $\Omega_v$, respectively. We also consider a finite training dataset $(\boldsymbol{f}_i, \boldsymbol{g}_i)_{i=1}^{N_{\text{train}}}$ of pairs of functions $\boldsymbol{g}_i = \mathcal{F}(\boldsymbol{f}_i)$. The unknown operator $\mathcal{F} : \mathcal{U} \to \mathcal{V}$ is to be approximated by a neural network $\mathsf{N} : \mathcal{U} \to \mathcal{V}$ trained by empirical risk minimization, as described in Equation (1).

**PCA-Net:** In order to construct the PCA-Net architecture, consider the eigen-decomposition of the empirical covariance operator of the training set of functions $(\boldsymbol{f}_i)_{i=1}^{N_{\text{train}}}$ and $(\boldsymbol{g}_i)_{i=1}^{N_{\text{train}}}$. This allows one to consider a truncated orthonormal PCA basis $(\boldsymbol{e}_{u,1}, \ldots, \boldsymbol{e}_{u,N_u})$ of $\mathcal{U}$, as well as a truncated PCA basis $(\boldsymbol{e}_{v,1}, \ldots, \boldsymbol{e}_{v,N_v})$ of $\mathcal{V}$.

The projection operator $\mathcal{A}_{\text{PCA}} : \mathcal{U} \to \mathbb{R}^{N_u}$ onto the PCA basis is defined as $\mathcal{A}_{\text{PCA}}(\boldsymbol{f}) = (\langle \boldsymbol{f}, \boldsymbol{e}_{u,1} \rangle, \ldots, \langle \boldsymbol{f}, \boldsymbol{e}_{u,N_u} \rangle)$. Similarly, define the linear operator $\mathcal{P}_{\text{PCA}} : \mathbb{R}^{N_v} \to \mathcal{V}$ as $\mathcal{P}_{\text{PCA}}(\boldsymbol{\beta}) = \beta_1 \boldsymbol{e}_{v,1} + \ldots + \beta_{N_v} \boldsymbol{e}_{v,N_v}$. With these definitions, the PCA-Net architecture is defined as

$$\mathsf{N}_{\text{PCA}} = \mathcal{P}_{\text{PCA}} \circ \mathsf{MLP} \circ \mathcal{A}_{\text{PCA}} \tag{17}$$

where $\mathsf{MLP} : \mathbb{R}^{N_u} \to \mathbb{R}^{N_v}$ is a standard multilayer perceptron, i.e., a fully $L$-layer connected feedforward neural network, with hidden layers of width $C$. Because the PCA bases are fixed, the computational cost of the PCA-Net architecture is independent of the discretization of the PDE. Furthermore, the PCA-Net method can readily be applied to PDE problems defined in irregular domains.

**POD-DeepONet:** The DeepONet architecture (Lu et al., 2021) encodes the input function space in the branch network and the domain of output functions in the trunk network, respectively. Specifically, for an input function, $\boldsymbol{f} \in \mathcal{U}$, the branch network $\mathcal{B} : \mathcal{U} \to \mathbb{R}^P$ takes the discretized input function $\boldsymbol{f}$ as input and outputs a vector $\mathcal{B}(\boldsymbol{f}) \in \mathbb{R}^P$. For $\boldsymbol{y} \in \Omega_v$, the trunk network $\mathcal{T} : \Omega_v \to \mathbb{R}^{P,d_{\text{out}}}$ takes the coordinates of the output function as input and outputs a vector $\mathcal{T}(\boldsymbol{y}) \in \mathbb{R}^{P,d_{\text{out}}}$. The vanilla unstacked DeepONet is then defined as

$$\mathsf{N}_{\text{DeepONet}}(\boldsymbol{f})(\boldsymbol{y}) = \sum_{i=1}^{P} \mathcal{B}_i(\boldsymbol{f}) \mathcal{T}_i(\boldsymbol{y}) + b_0 = \mathcal{B}(\boldsymbol{f})^\top \mathcal{T}(\boldsymbol{y}) + b_0, \tag{18}$$

for a bias vector $b_0 \in \mathbb{R}^{d_{\text{out}}}$. The architecture of the branch network depends on the structure of the problem and is typically chosen as a CNN or an RNN or an MLP. The trunk network is typically a standard MLP.

The POD-DeepONet architecture, an extension of the vanilla DeepONet, was proposed in Lu et al. (2022). The POD-DeepONet use precomputed proper orthogonal decomposition (POD) basis functions instead of a trunk net. It is defined as

$$\mathsf{N}_{\text{PODD}}(\boldsymbol{f}) = \sum_{i=1}^{P} \mathcal{B}_i(\boldsymbol{f}) \, \boldsymbol{e}_i + \boldsymbol{e}_0, \tag{19}$$

where $\mathcal{B} : \mathcal{U} \to \mathbb{R}^P$ denotes a standard branch net as used in the vanilla DeepOnet architecture and $(\boldsymbol{e}_1, \ldots, \boldsymbol{e}_P)$ are $P \geq 1$ precomputed basis functions of $\mathcal{V}$ that are generated from training data and $\boldsymbol{e}_0$ is the mean function. The experiments presented in Lu et al. (2022) suggest that the POD-DeepONet architecture outperforms the vanilla DeepONet architecture for a wide class of PDE problems. In our simulations, following Lu et al. (2022), the combination of CNN with $C$ channels and MLP of width $K$ is used as the branch network.

**Fourier neural operator (FNO)** For an input function $\boldsymbol{f} \in \mathcal{U}$, the FNO architecture (Li et al., 2021) is defined as

$$\mathsf{N}_{\text{FNO}}(\boldsymbol{f}) := \mathcal{Q} \circ \mathcal{L}_L \circ \cdots \mathcal{L}_1 \circ \mathcal{P}(\boldsymbol{f}). \tag{20}$$

The lift operator $\mathcal{P}$ and projection operator $\mathcal{Q}$ are defined as

$$\mathcal{P}(\boldsymbol{f}) = \boldsymbol{P}\boldsymbol{f} + \boldsymbol{b}_{\mathcal{P}}, \qquad \text{and} \qquad \mathcal{Q}(\boldsymbol{f}_L) = \boldsymbol{Q}\boldsymbol{f}_L + \boldsymbol{b}_{\mathcal{Q}},$$

with vectors $\boldsymbol{b}_{\mathcal{P}} \in \mathbb{R}^C$ and $\boldsymbol{b}_{\mathcal{Q}} \in \mathbb{R}^{d_{\text{out}}}$ and matrices $\boldsymbol{P} \in \mathbb{R}^{C,d_{\text{in}}}$ and $\boldsymbol{Q} \in \mathbb{R}^{d_{\text{out}},C}$. The Fourier layers $\mathcal{L}_1, \ldots, \mathcal{L}_L$ in Equation (20) are defined as

$$\boldsymbol{f}_l(\boldsymbol{x}) = \mathcal{L}_l(\boldsymbol{f}_{l-1})(\boldsymbol{x}) = \sigma\left(\boldsymbol{W}_l(\boldsymbol{f}_{l-1}(\boldsymbol{x})) + \mathscr{F}^{-1}\left(\boldsymbol{R}_l \cdot \mathscr{F}(\boldsymbol{f}_{l-1})\right)(\boldsymbol{x})\right) \tag{21}$$

for matrix $\boldsymbol{W}_l \in \mathbb{R}^{C,C}$ and $\boldsymbol{R}_l(\boldsymbol{s}) \in \mathbb{C}^{C \times C}$ for each frequency $\boldsymbol{s}$. The notations $\mathscr{F}$ and $\mathscr{F}^{-1}$ denote the Fourier transform and its inverse. In Equation (21), $\sigma$ denotes a nonlinear activation function except in the last layer $\mathcal{F}_L$ where $\sigma$ is the identity function.

## 5.2 Performance comparisons

This section evaluates the performance of GIT-Net on the PDE problems introduced in Section 3 when compared to the baseline methods described in Section 5.1.

**Configurations**  To study the impact of the hyperparameters, the following settings were considered:

$(H)$
> FNO: $C \in \{2, 4, 8, 16, 32\}$;
>
> PCA-Net: $K \in \{16, 64, 128, 256, 512, 1024, 2048, 8096\}$
>
> POD-DeepOnet: $(C, K) \in \{(2, 16), (4, 64), (8, 128), (16, 256), (32, 512)\}$;
>
> GIT-Net: $C \in \{2, 4, 8, 16, 32\}, K \in \{16, 64, 128, 256, 512\}$.

Following the setup of the Fourier Neural Operator (FNO) in de Hoop et al. (2022), we used twelve Fourier modes and three Fourier Neural Layers ($L = 3$ in (20)). For the PCA-Net, as in de Hoop et al. (2022), we use four internal layers (4-layer MLP in (17)). For the POD-DeepONet, following the recommendations of Lu et al. (2022), we use a two-dimensional convolutional neural network and fully connected layers as branch nets for the two-dimensional problems, and a fully connected network as the branch net for one-dimensional problems. If the input of a two-dimensional PDE is a one-dimensional boundary condition, it is first expanded to two dimensions before being fed into the convolutional neural network. We test various pairs of the number of channels $C$ of the convolutional layers and the width of the fully connected layers $K$. The activation function used is the GELU function in FNO and the RELU function in PCA-Net and POD-DeepONet.

We employed the aforementioned configurations to assess and compare the neural network operators based on their test error, error profile, and evaluation cost. These metrics collectively evaluate various aspects of the operators' performance.

**Test error**  Figure 9 compares the relative test errors defined in Equation (16). For this comparison, we only show results corresponding to the hyperparameters $(C, K) \in \{(2, 16), (4, 64), (8, 128), (16, 256), (32, 512)\}$ for GIT-Net as representatives. In the large data regime $N_{\text{train}} = 20,000$, GIT-Net consistently achieves the smallest test error for all tested PDE problems. Furthermore, the FNO and GIT-Net architecture outperforms other methods for PDE problems defined on rectangular grids. For these simple geometries and in the small data regimes, FNO performs slightly better than GIT-Net, with a lower test error and less overfitting. However, FNO does not behave as well for more complex geometries. In the structural mechanics equation, FNO obtains similar test errors to POD-DeepONet and GIT-Net. For the Poisson equation, the test error of FNO is significantly larger than that of the other methods. The PCA-Net and POD-DeepONet architectures achieve similar test errors for problems defined on rectangular grids. In these cases, PCA-Net can be seen as a POD-DeepONet with a fully connected neural network as a branch net. However, for problems defined on more complex geometries, the PCA-Net performs better for the structural mechanics equation, but worse for the Poisson equation when compared to the POD-DeepONet. Note that when $C = 1$, the PCA-Net and GIT-Net architectures are equivalent. This remark is confirmed by Figure 9 which indicates that when $C = 2$ and $K$ is small (e.g. 16), the performance of the GIT-Net and PCA-Net are similar.

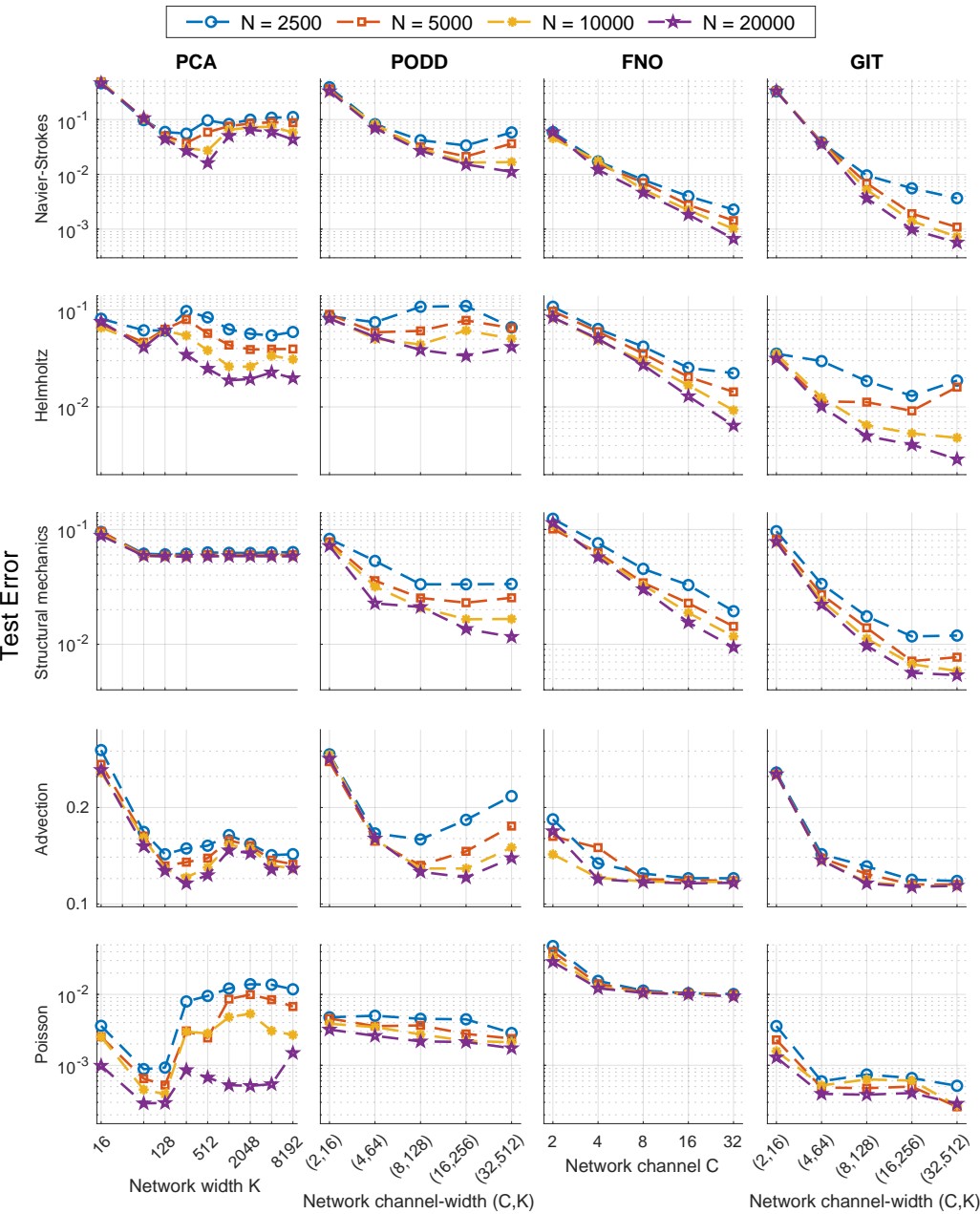

Figure 9: Test error of neural network operators for all PDE problems.

**Error profiles** Figure 9 illustrates the average prediction error on the test data. To gain further insights, the statistical distribution of the test errors and profiles of median-error cases and worst-error cases are provided. To demonstrate the best performance that these network operators can achieve, we selected the predictions and error profiles corresponding to the hyperparameters $C$ and $K$ that achieve the minimum

test error on $N_{\text{train}} = 20,000$ data samples. The procedure for selecting hyperparameters and cases which is denoted by (S) is as follows.

1. **Train network:** For each PDE problem, train the networks of architecture PCA-Net, POD-DeepOnet, FNO, and GIT-Net with $20,000$ training data, where the used hyperparameter sets in (H) are denoted by $\{h_k^{\text{PCA}}\}_{k=1}^9$, $\{h_k^{\text{PODD}}\}_{k=1}^5$, $\{h_k^{\text{FNO}}\}_{k=1}^5$, and $\{h_k^{\text{GIT}}\}_{k=1}^{25}$, respectively.

2. **Choose hyperparameters with minimum test errors:** Test these architectures with another $20,000$ test data to get the test error $\{e_k^{\text{PCA}}\}_{k=1}^9$, $\{e_k^{\text{PODD}}\}_{k=1}^5$, $\{e_k^{\text{FNO}}\}_{k=1}^5$, and $\{e_k^{\text{GIT}}\}_{k=1}^{25}$ for all tested hyperparameters. For each architecture, we choose the hyperparameter sets that minimize the test errors which are denoted by $h_{k_1}^{\text{PCA}}$, $h_{k_2}^{\text{PODD}}$, $h_{k_3}^{\text{FNO}}$, and $h_{k_4}^{\text{GIT}}$. Here, the test error means the average over all test data.

3. **Choose input-output image pairs and their error image:** For these architectures with selected hyperparameters $h_{k_1}^{\text{PCA}}$, $h_{k_2}^{\text{PODD}}$, $h_{k_3}^{\text{FNO}}$, and $h_{k_4}^{\text{GIT}}$, show the inputs, outputs, and predictions with largest test error and median test error among $20000$ test data.

We choose hyperparameters for all neural network operators according to Step 2 above and show their test error distribution histograms on 20000 test data in Figure 10, 13, 16, 19 and 22. Then according to Step 3, the images of median-error and worst-error cases, including their input, output, prediction, and error images, are shown in Figure 12, 15, 18, 21 and 24. The comparison of error profiles for the same sample are shown in Figure 11, 14, 17, 20 and 23.

Figure 10 shows a histogram of test errors of selected models on 20000 test data for Navier-Stokes equation (11), where used hyperparameters of each neural network operator are

$$h_{k_1}^{\text{PCA}} = \{K = 512\}, h_{k_2}^{\text{PODD}} = \{(C, K) = (32, 512)\}, h_{k_3}^{\text{FNO}} = \{C = 32\}, h_{k_4}^{\text{GIT}} = \{(C, K) = (512)\}.$$

Figure 11 shows a sample of test error, including input, output, and predictions of all neural network operators and corresponding error images. Figure 12 shows the median-error cases and worst-error cases. It can be seen from the histogram 10 that GIT-Net and FNO are significantly better than the other two methods. At the same time, the upper limit of the test error of GIT-Net is smaller than that of FNO, but the lower limit is similar. The test error interval of GIT-Net is more concentrated than that of FNO. From the error profiles of the median-error cases, it can be seen that FNO and GIT-Net achieve much smaller test errors than the other methods. Moreover, the error profiles of FNO and GIT-Net exhibit analogous structures for the worst error cases, indicating the presence of similar function approximators in both scenarios. This observation is further supported by the example shown in Figure 11.

Figure 13 shows a histogram of test errors of selected models on 20000 test data for Helmholtz equation (12), where used hyperparameters of each neural network operator are

$$h_{k_1}^{\text{PCA}} = \{K = 512\}, h_{k_2}^{\text{PODD}} = \{(C, K) = (16, 256)\}, h_{k_3}^{\text{FNO}} = \{C = 32\}, h_{k_4}^{\text{GIT}} = \{(C, K) = (32, 512)\}.$$

Figure 14 shows an example and Figure 15 shows the median-error cases and worst-error cases. From the histogram, we can see that the lower limit of the test error of GIT-Net is slightly smaller than that of FNO, and the upper limit is almost the same, but the test error interval of FNO is more concentrated. Not only in Figure 13 but also in Figure 15, FNO and GIT-Net showed significantly better predictions. Meanwhile, their error images of median-error cases and worst-error cases had similar textures and intensity.

Figure 16 shows a histogram of test errors of the selected model on 20000 test data for structural mechanics problem (13), where used hyperparameters of each neural network operator are

$$h_{k_1}^{\text{PCA}} = \{K = 256\}, h_{k_2}^{\text{PODD}} = \{(C, K) = (32, 512)\}, h_{k_3}^{\text{FNO}} = \{C = 32\}, h_{k_4}^{\text{GIT}} = \{(C, K) = (32, 512)\}.$$

Figure 17 shows an example and Figure 18 shows the median-error cases and worst-error cases. From the histogram, the test error ranges of POD-DeepOnet and FNO are similar, but FNO is more concentrated.

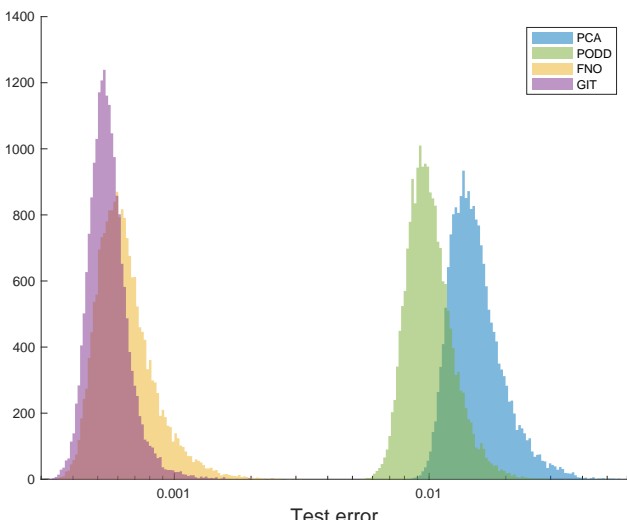

Figure 10: Histogram of test errors of Navier-Stokes equation

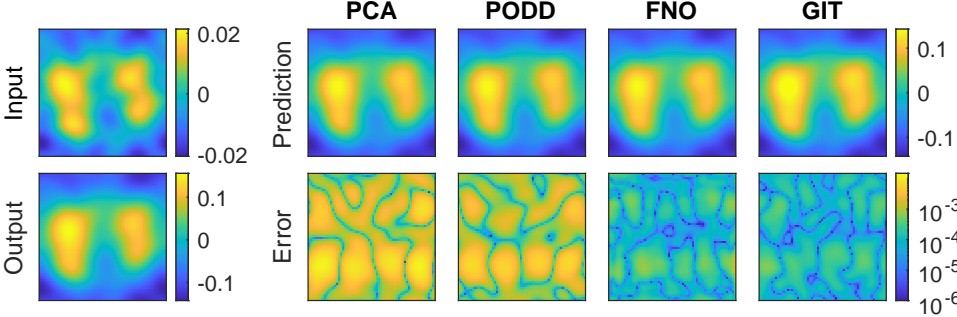

Figure 11: An example of Navier-Stokes, including input, output, predictions, and error images of neural network operators.

GIT-Net has the smallest upper and lower bounds, and PCA-Net's is the largest one. In the median-error case, the error profiles of the GIT-Net and POD-DeepONet have similar intensities, while the error profile of the FNO exhibits more severe oscillation structures, particularly near the hole, which is likely due to the interpolation of triangular meshes onto rectangular meshes. In the worst-error case, the PCA-Net obtains the largest error, while the intensity of the error image of GIT-Net is slightly smaller than FNO and POD-DeepONet. The error profiles of all methods show the most violent oscillations near the top boundary, where the traction force is applied, leading to more complex changes in stress in the output space.

Figure 19 shows a histogram of test errors of the selected models on 20000 test data for advection equation (14), where used hyperparameters of each neural network operator are

$$h_{k_1}^{\text{PCA}} = \{K = 256\}, h_{k_2}^{\text{PODD}} = \{(C, K) = (16, 256)\}, h_{k_3}^{\text{FNO}} = \{C = 16\}, h_{k_4}^{\text{GIT}} = \{(C, K) = (16, 256)\}.$$

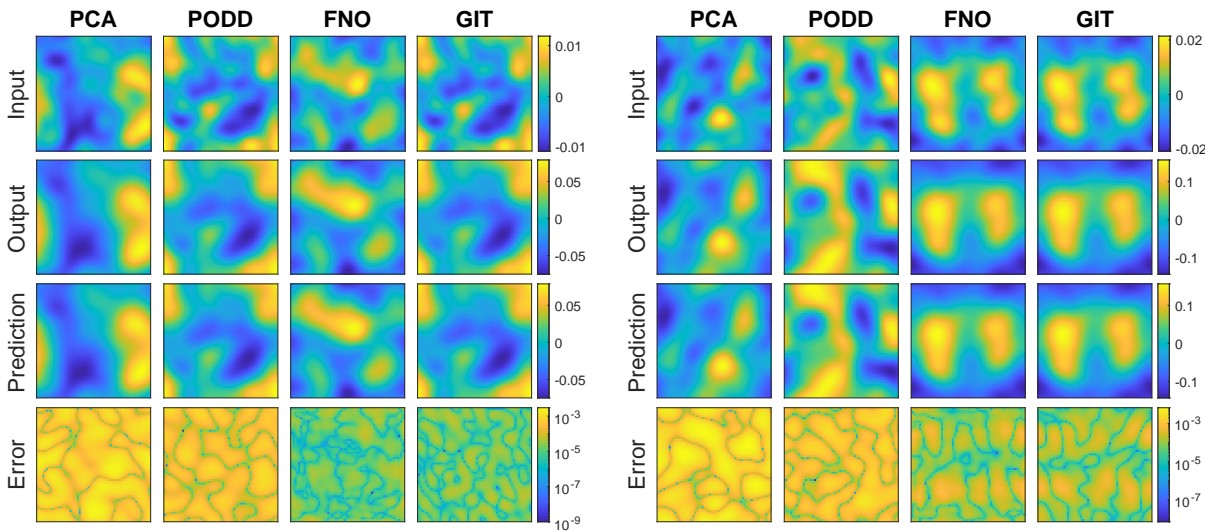

Figure 12: Input, output, and prediction of Navier-Stokes equation using the hyperparameters selected by (S) on 20000 testing data for each neural network. Left: the case with median test error. Right: the case with the largest test error.

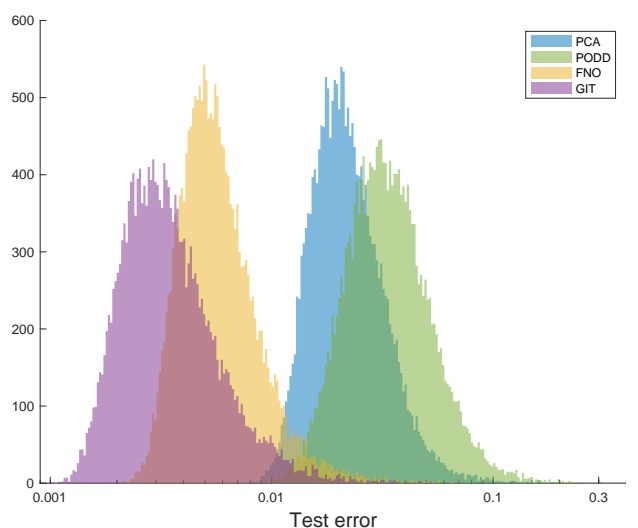

Figure 13: Histogram of test errors of Helmholtz equation

Figure 20 shows an example and Figure 21 shows the median-error cases and worst-error cases. As can be seen from the histogram, the mean values and ranges of the test errors of all neural network operators are similar, but the distribution of FNO is more concentrated. From the median-error case, it can be seen that all operators produce visually good predictions, including sharp jump points. In the worst-error cases, GIT-Net is observed to predict jump points much more accurately than the other operators. This is consistent with the observation from the histogram that the test error upper limit of GIT-Net is smaller than that of FNO. This may be due to the fact that FNO is more inclined to fit smooth, low-frequency periodic functions, and

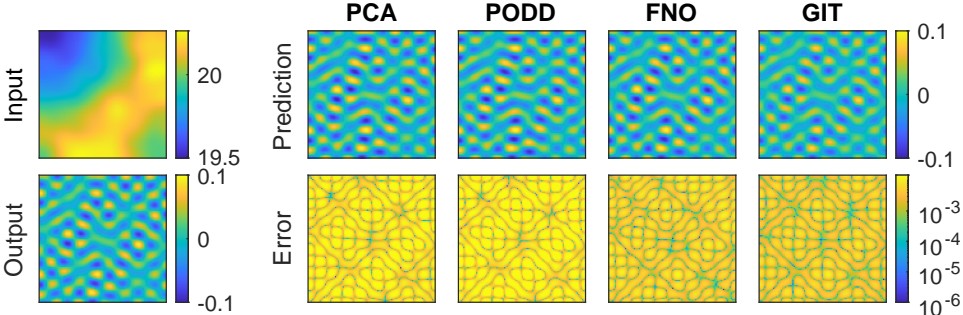

Figure 14: An example of Helmholtz equation, including input, output, predictions, and error images of neural network operators.

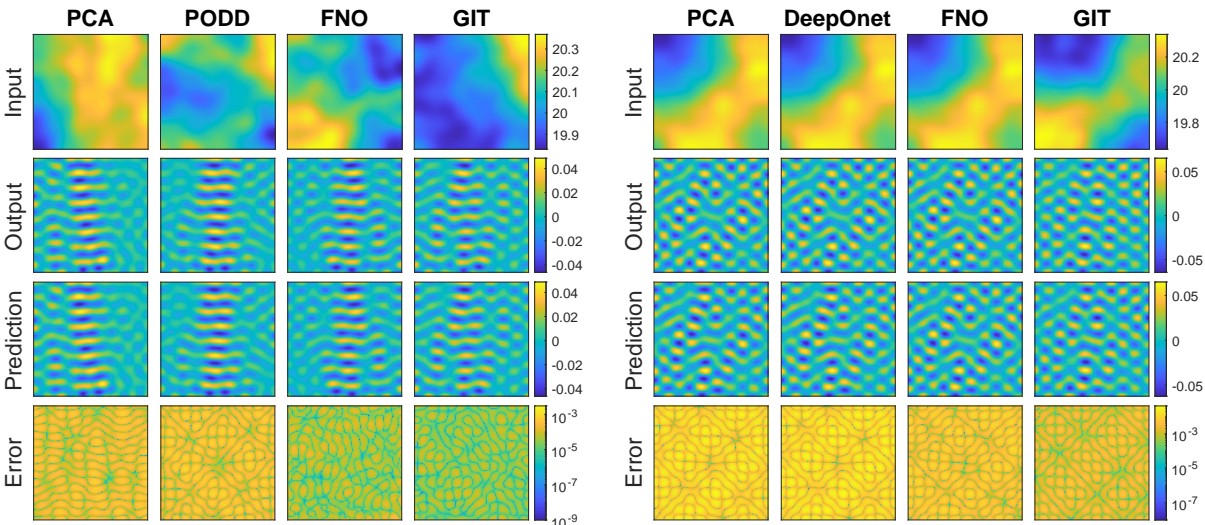

Figure 15: Input, output, and prediction of Helmholtz equation using the hyperparameters selected by (S) on 20000 testing data for each neural network. Left: the case with median test error. Right: the case with the largest test error.

thus its ability to fit the jump points of piecewise linear functions is limited by the number of Fourier modes used.

Figure 22 shows a histogram of test errors of selected models on 20000 test data for Poisson equation (15), where used hyperparameters of each neural network operator are

$$h_{k_1}^{\text{PCA}} = \{K = 64\}, h_{k_2}^{\text{PODD}} = \{(C, K) = (16, 256)\}, h_{k_3}^{\text{FNO}} = \{C = 32\}, h_{k_4}^{\text{GIT}} = \{(C, K) = (32, 512)\}.$$

Figure 23 shows an example and Figure 24 shows the median-error cases and worst-error cases. Since the test error ranges of PCA-Net and GIT-Net almost coincide, in order to more clearly represent the difference

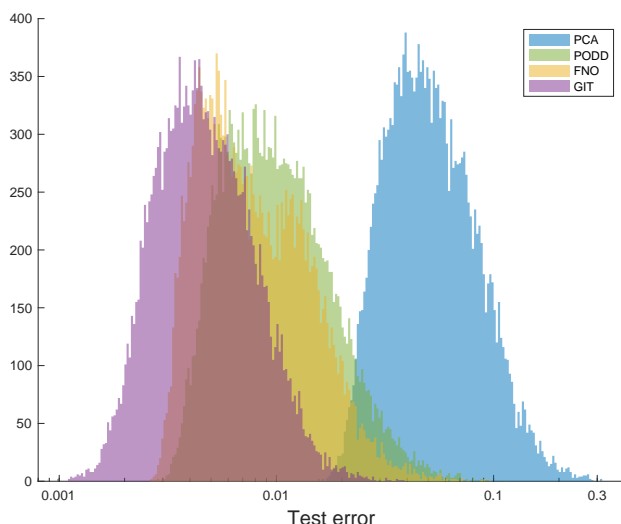

Figure 16: Histogram of test errors of structural mechanics equation

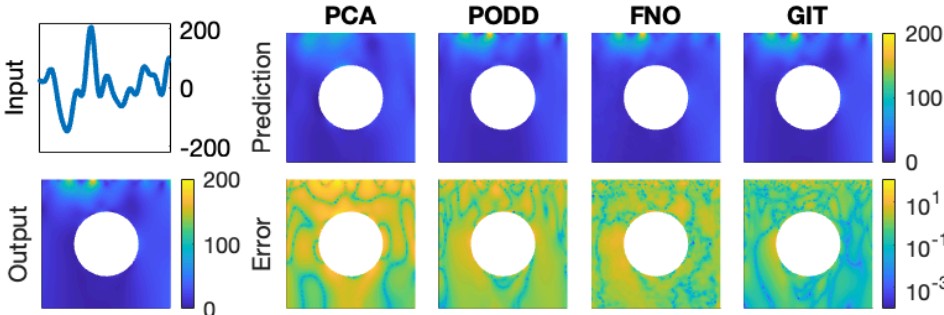

Figure 17: An example of structural mechanics equation, including input, output, predictions, and error images of neural network operators.

between the two, the vertical axis in the histogram is set to a logarithmic scale. It can be observed that although the mean of test error of GIT-Net is very close to that of PCA-Net, GIT-Net is more concentrated, and its upper bound is much smaller than that of PCA-Net. In the median-error case, the PCA-Net and GIT-Net achieve better predictions than the other methods. The areas with large errors tend to be where the output values are large. In the worst-error case, GIT-Net performs better than the other methods. Although the PCA-Net method obtains a smaller average of test errors, it performs slightly worse than the POD-DeepOnet method in the worst-error case, which indicates that the test error variance of PCA-Net is larger, which is consistent with the histogram. The error profile obtained by the FNO method shows severe oscillations, which is likely due to interpolation error.

**Evaluation cost**    The approximate PDE operators discussed in this text are especially useful in situations such as Bayesian Inverse Problems when it is necessary to repeatedly compute solutions to PDEs. As a result, their evaluation costs are an important consideration. In de Hoop et al. (2022), a cost analysis based on

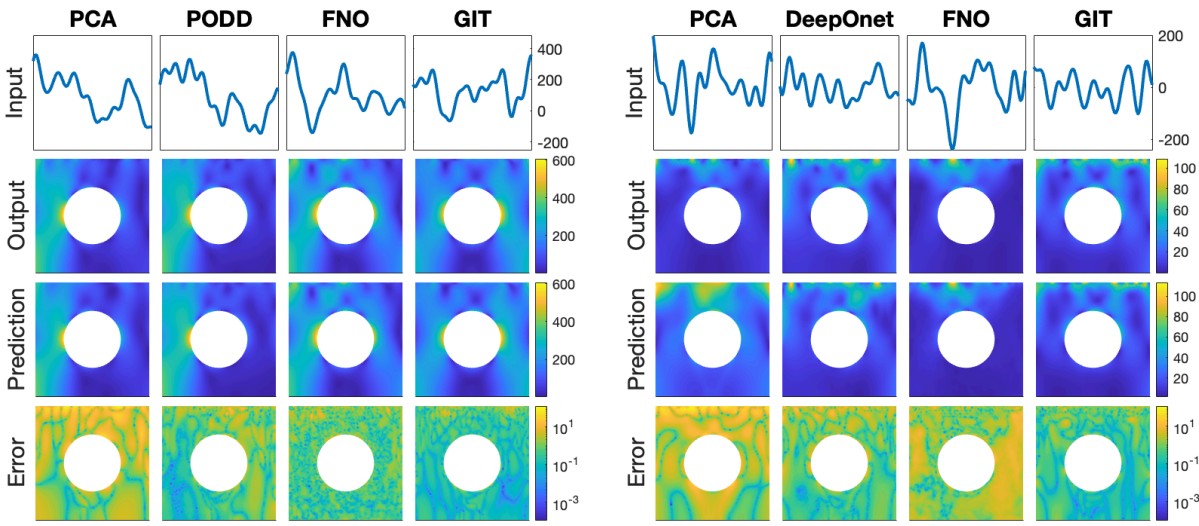

Figure 18: Input, output, and prediction of structural mechanics equation using the hyperparameters selected by (S) on 20000 testing data for each neural network. Left: the case with median test error. Right: the case with the largest test error.

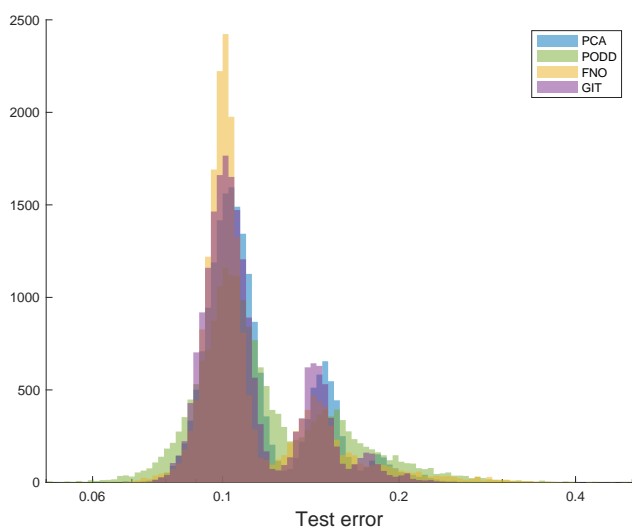

Figure 19: Histogram of test errors of advection equation

floating-point operations was provided. This analysis showed that, assuming the number of sampling points for the input and output functions is $\mathcal{O}(N_p)$, the evaluation costs for PCA-Net and FNO scale as $\mathcal{O}(N_p + K^2)$ and $\mathcal{O}(CN_p \log(N_p) + N_p C^2)$, respectively. The cost scaling of POD-DeepONet is $\mathcal{O}(N_p C^2 + CKN_p + N_p)$ for two-dimensional problems and $\mathcal{O}(N_p + K^2)$ for one-dimensional problems. In contrast, the cost scaling of GIT-Net is $\mathcal{O}(N_p + CK(C + K))$. Figure 25 reports the evaluation costs as a function of the test errors for all of the neural network operators considered in this study and for four different amounts of training data.

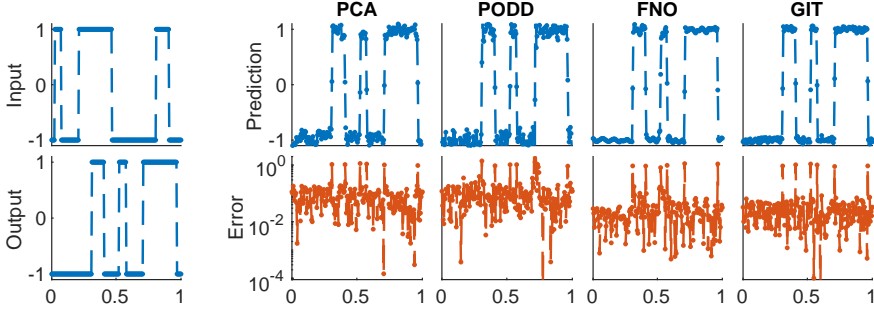

Figure 20: An example of advection equation, including input, output, predictions, and error images of neural network operators.

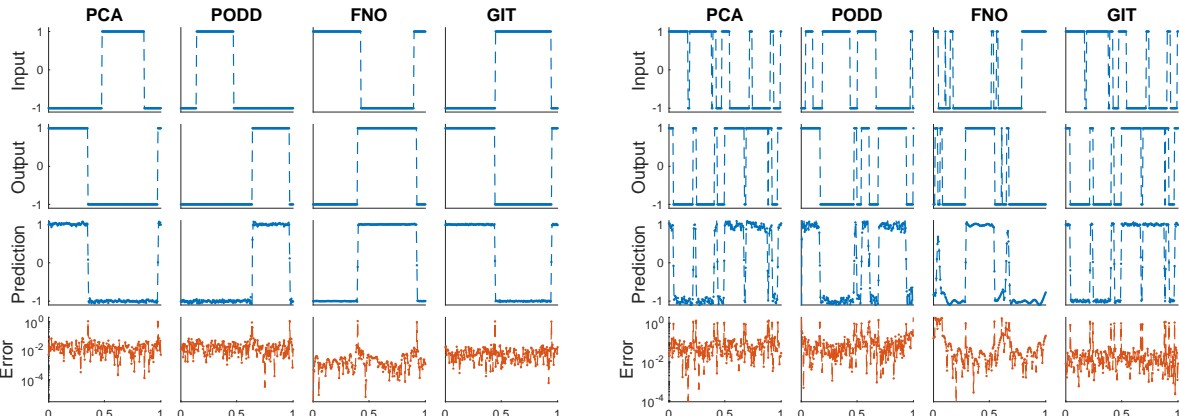

Figure 21: Input, output, and prediction of advection equation using the hyperparameters selected by (S) on 20000 testing data for each neural network. Left: the case with median test error. Right: the case with the largest test error.

For the Navier-Stokes equation, GIT-Net and FNO achieve the best performance and similar test errors with similar evaluation costs when a sufficient amount of training data is used (e.g. $N_{\text{train}} = 20,000$). However, when the amount of training data is small (e.g. $N_{\text{train}} = 2500$), GIT-Net exhibits more pronounced overfitting when compared to FNO. PCA-Net and POD-DeepONet show even more severe overfitting in this regime. For the Helmholtz equation and structural mechanics problems, GIT-Net achieves much smaller test errors than the other methods at the same evaluation cost. For the advection equation, FNO can achieve smaller test errors than the other methods at low evaluation costs, but GIT-Net performs better as the cost increases. For the Poisson equation, FNO performs the worst, with the largest evaluation cost and largest error. Overall, FNO and GIT-Net outperform PCA-Net and POD-DeepONet on problems defined on rectangular grids and can achieve similar test errors. However, FNO exhibits less overfitting on the Navier-Stokes equation with a small amount of training data. GIT-Net achieves similar or better test errors than FNO at a similar or lower evaluation cost. For problems defined on complex geometries, the FNO architecture exhibits much larger test errors than the GIT-Net approach.

## 6 Conclusion

We present GIT-Net, a neural network operator designed for the approximation of partial differential equation (PDE) operators. Our data-driven approach learns an integral transform from paired input-output functions.

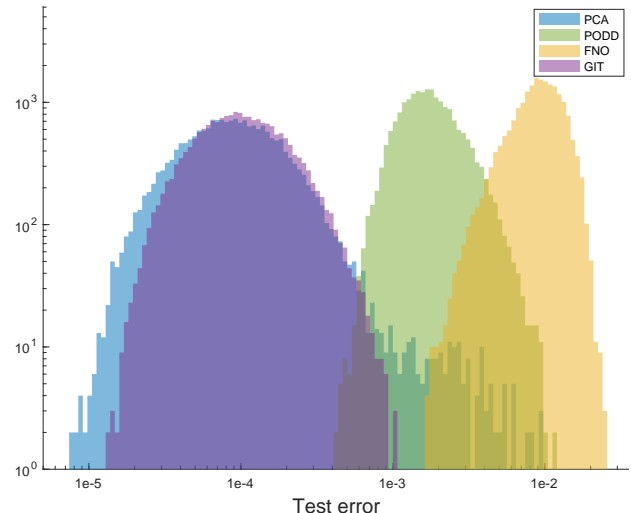

Figure 22: Histogram of test errors of Poisson equation

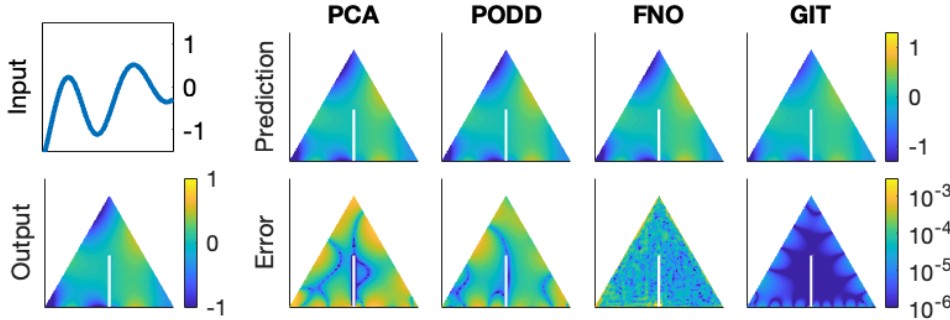

Figure 23: An example of Poisson equation, including input, output, predictions, and error images of neural network operators.

GIT-Net exhibits robustness to mesh discretizations and can be readily implemented for PDE problems on complex geometries or when the input and output functions are defined on disparate domains.

We demonstrate the efficacy of GIT-Net across a variety of PDE problems of varying dimensions and geometries. When compared with recently proposed operator learning methods, including PCA-Net, POD-DeepONet, and FNO, GIT-Net consistently delivers the lowest test error in the large data regime. For PDEs defined on rectangular grids, GIT-Net and FNO typically reach comparable accuracies, although GIT-Net typically operates at lower computational costs. Our experiments suggest that on more complex geometries, FNO is not competitive when compared to GIt-Net. In terms of scalability with respect to the number of sampling points $N_p$, the complexity of FNO and GIT-Net are $\mathcal{O}(CN_p \log(N_p) + N_p C^2)$ and $\mathcal{O}(N_p + C^2 K + CK^2)$, respectively. These results suggest that GIT-Net has favorable properties for learning

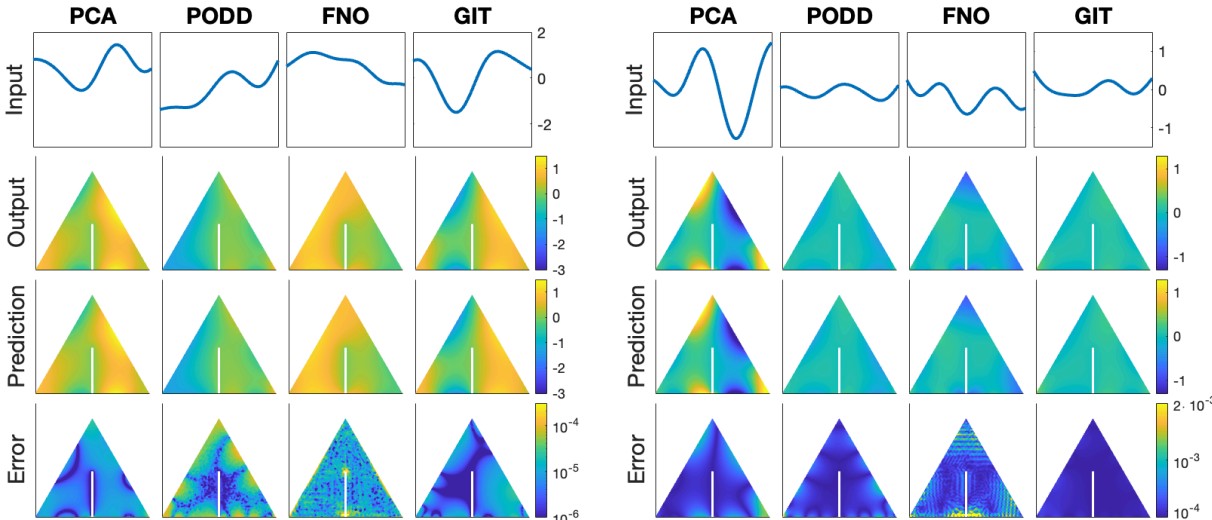

Figure 24: Input, output, and prediction of Poisson equation using the hyperparameters selected by (S) on 20000 testing data for each neural network. Left: the case with median test error. Right: the case with the largest test error.

PDE operators in a wide range of settings. As a part of ongoing work, we are establishing a theoretical framework to elucidate these empirical observations.

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

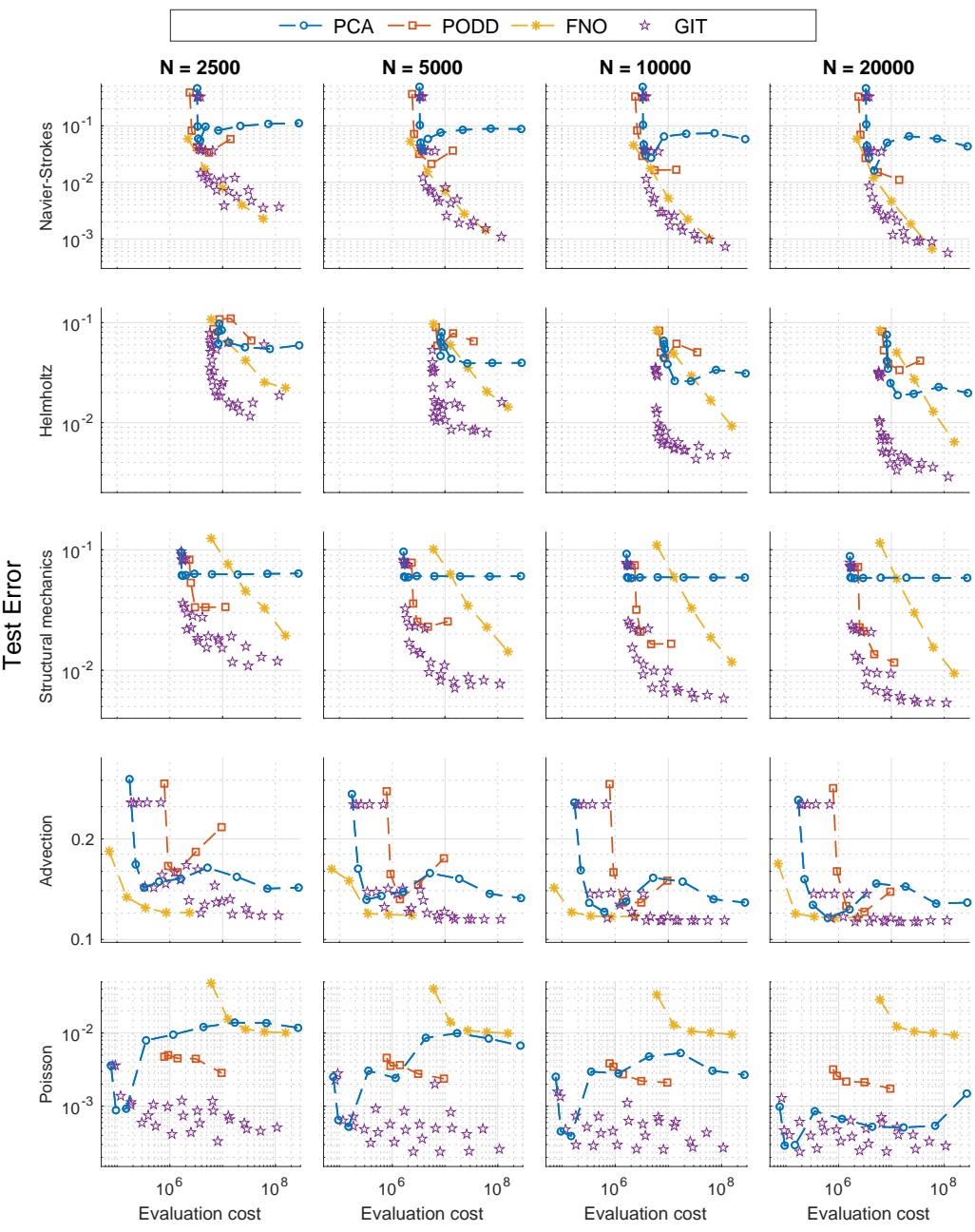

Figure 25: Evaluation cost vs test error of all methods for all problems.

Shuhao Cao. Choose a transformer: Fourier or galerkin. *Advances in neural information processing systems*, 34:24924–24940, 2021.

François Chollet. Xception: Deep learning with depthwise separable convolutions. In *Proceedings of the IEEE conference on computer vision and pattern recognition*, pp. 1251–1258, 2017.

Ingrid Daubechies, Ronald DeVore, Simon Foucart, Boris Hanin, and Guergana Petrova. Nonlinear approximation and (deep) relu networks. *Constructive Approximation*, 55(1):127–172, 2022.

Maarten de Hoop, Daniel Zhengyu Huang, Elizabeth Qian, and Andrew M. Stuart. The cost-accuracy trade-off in operator learning with neural networks. *Journal of Machine Learning*, 1(3):299–341, 2022. ISSN 2790-2048. doi: https://doi.org/10.4208/jml.220509. URL http://global-sci.org/intro/article_detail/jml/21030.html.

Weinan E and Bing Yu. The deep ritz method: a deep learning-based numerical algorithm for solving variational problems. *Communications in Mathematics and Statistics*, 6(1):1–12, 2018.

Weinan E, Chao Ma, and Lei Wu. Barron spaces and the compositional function spaces for neural network models. *arXiv preprint arXiv:1906.08039*, 2019.

Tarek A El Moselhy and Youssef M Marzouk. Bayesian inference with optimal maps. *Journal of Computational Physics*, 231(23):7815–7850, 2012.

Robert Eymard, Thierry Gallouët, and Raphaèle Herbin. Finite volume methods. *Handbook of numerical analysis*, 7:713–1018, 2000.

Xiaoxiao Guo, Wei Li, and Francesco Iorio. Convolutional neural networks for steady flow approximation. In *Proceedings of the 22nd ACM SIGKDD international conference on knowledge discovery and data mining*, pp. 481–490, 2016.

Gaurav Gupta, Xiongye Xiao, and Paul Bogdan. Multiwavelet-based operator learning for differential equations. *Advances in neural information processing systems*, 34:24048–24062, 2021.

Nathan Halko, Per-Gunnar Martinsson, and Joel A Tropp. Finding structure with randomness: Probabilistic algorithms for constructing approximate matrix decompositions. *SIAM review*, 53(2):217–288, 2011.

Juncai He, Lin Li, Jinchao Xu, and Chunyue Zheng. Relu deep neural networks and linear finite elements. *Journal of Computational Mathematic*, 38(3):502–527, 2020.

Dan Hendrycks and Kevin Gimpel. Bridging nonlinearities and stochastic regularizers with gaussian error linear units. *CoRR*, abs/1606.08415, 2016. URL http://arxiv.org/abs/1606.08415.

Jan S Hesthaven and Stefano Ubbiali. Non-intrusive reduced order modeling of nonlinear problems using neural networks. *Journal of Computational Physics*, 363:55–78, 2018.

Xiang Huang, Zhanhong Ye, Hongsheng Liu, Shi Ji, Zidong Wang, Kang Yang, Yang Li, Min Wang, Haotian Chu, Fan Yu, et al. Meta-auto-decoder for solving parametric partial differential equations. *Advances in Neural Information Processing Systems*, 35:23426–23438, 2022.

George Em Karniadakis, Ioannis G Kevrekidis, Lu Lu, Paris Perdikaris, Sifan Wang, and Liu Yang. Physics-informed machine learning. *Nature Reviews Physics*, 3(6):422–440, 2021.

Yuehaw Khoo, Jianfeng Lu, and Lexing Ying. Solving parametric pde problems with artificial neural networks. *European Journal of Applied Mathematics*, 32(3):421–435, 2021.

Diederik P. Kingma and Jimmy Ba. Adam: A method for stochastic optimization. In Yoshua Bengio and Yann LeCun (eds.), *3rd International Conference on Learning Representations, ICLR 2015, San Diego, CA, USA, May 7-9, 2015, Conference Track Proceedings*, 2015. URL http://arxiv.org/abs/1412.6980.

Georgios Kissas, Jacob H Seidman, Leonardo Ferreira Guilhoto, Victor M Preciado, George J Pappas, and Paris Perdikaris. Learning operators with coupled attention. *Journal of Machine Learning Research*, 23 (215):1–63, 2022.

Nikola Kovachki, Samuel Lanthaler, and Siddhartha Mishra. On universal approximation and error bounds for fourier neural operators. *Journal of Machine Learning Research*, 22:Art–No, 2021.

Zijie Li, Kazem Meidani, and Amir Barati Farimani. Transformer for partial differential equations' operator learning. *Transactions on Machine Learning Research*, 2023. ISSN 2835-8856. URL `https://openreview.net/forum?id=EPPqt3uERT`.

Zongyi Li, Nikola Borislavov Kovachki, Kamyar Azizzadenesheli, Burigede liu, Kaushik Bhattacharya, Andrew Stuart, and Anima Anandkumar. Fourier neural operator for parametric partial differential equations. In *International Conference on Learning Representations*, 2021. URL `https://openreview.net/forum?id=c8P9NQVtmnO`.

Tadeusz Liszka and Janusz Orkisz. The finite difference method at arbitrary irregular grids and its application in applied mechanics. *Computers & Structures*, 11(1-2):83–95, 1980.

Zichao Long, Yiping Lu, Xianzhong Ma, and Bin Dong. Pde-net: Learning pdes from data. In *International Conference on Machine Learning*, pp. 3208–3216. PMLR, 2018.

Zichao Long, Yiping Lu, and Bin Dong. Pde-net 2.0: Learning pdes from data with a numeric-symbolic hybrid deep network. *Journal of Computational Physics*, 399:108925, 2019.

Lu Lu, Pengzhan Jin, Guofei Pang, Zhongqiang Zhang, and George Em Karniadakis. Learning nonlinear operators via DeepONet based on the universal approximation theorem of operators. *Nature Machine Intelligence*, 3(3):218–229, 2021. ISSN 25225839. doi: 10.1038/s42256-021-00302-5.

Lu Lu, Xuhui Meng, Shengze Cai, Zhiping Mao, Somdatta Goswami, Zhongqiang Zhang, and George Em Karniadakis. A comprehensive and fair comparison of two neural operators (with practical extensions) based on fair data. *Computer Methods in Applied Mechanics and Engineering*, 393:114778, 2022.

Per-Gunnar Martinsson and Joel A Tropp. Randomized numerical linear algebra: Foundations and algorithms. *Acta Numerica*, 29:403–572, 2020.

Yong Zheng Ong, Zuowei Shen, and Haizhao Yang. Integral autoencoder network for discretization-invariant learning. *Journal of Machine Learning Research*, 23(286):1–45, 2022.

Shaowu Pan and Karthik Duraisamy. Physics-informed probabilistic learning of linear embeddings of nonlinear dynamics with guaranteed stability. *SIAM Journal on Applied Dynamical Systems*, 19(1):480–509, 2020.

Adam Paszke, Sam Gross, Soumith Chintala, Gregory Chanan, Edward Yang, Zachary DeVito, Zeming Lin, Alban Desmaison, Luca Antiga, and Adam Lerer. Automatic differentiation in pytorch. 2017.

Maziar Raissi, Paris Perdikaris, and George E Karniadakis. Physics-informed neural networks: A deep learning framework for solving forward and inverse problems involving nonlinear partial differential equations. *Journal of Computational physics*, 378:686–707, 2019.

David E Rumelhart, Geoffrey E Hinton, and Ronald J Williams. Learning internal representations by error propagation. Technical report, California Univ San Diego La Jolla Inst for Cognitive Science, 1985.

Yeonjong Shin, Jerome Darbon, and George Em Karniadakis. On the convergence of physics informed neural networks for linear second-order elliptic and parabolic type pdes. *Communications in Computational Physics*, 28(5):2042–2074, 2020. ISSN 1991-7120. doi: https://doi.org/10.4208/cicp.OA-2020-0193. URL `http://global-sci.org/intro/article_detail/cicp/18404.html`.

Justin Sirignano and Konstantinos Spiliopoulos. DGM: A deep learning algorithm for solving partial differential equations. *Journal of computational physics*, 375:1339–1364, 2018.

William S Slaughter. *The linearized theory of elasticity*. Springer Science & Business Media, 2012.

Jonathan D Smith, Kamyar Azizzadenesheli, and Zachary E Ross. Eikonet: Solving the eikonal equation with deep neural networks. *IEEE Transactions on Geoscience and Remote Sensing*, 59(12):10685–10696, 2020.

Ralph C Smith. *Uncertainty quantification: theory, implementation, and applications*, volume 12. Siam, 2013.

Andrew M Stuart. Inverse problems: a bayesian perspective. *Acta numerica*, 19:451–559, 2010.

Fredi Tröltzsch. *Optimal control of partial differential equations: theory, methods, and applications*, volume 112. American Mathematical Soc., 2010.

Yaohua Zang, Gang Bao, Xiaojing Ye, and Haomin Zhou. Weak adversarial networks for high-dimensional partial differential equations. *Journal of Computational Physics*, 411:109409, 2020.

Yinhao Zhu and Nicholas Zabaras. Bayesian deep convolutional encoder–decoder networks for surrogate modeling and uncertainty quantification. *Journal of Computational Physics*, 366:415–447, 2018.

## A    The effect of coordinates of grid as input in FNO method

Figure 26 compares the test error for FNO with and without grid coordinates as input. It is found that the two results are similar except for advection. For the advection equation, the with-grid case obtains smaller test error in most hyperparameters but except for $C = 2$ with 2500 training data, $C = 2, 4$ for 5000 training data, and $C = 2$ for 20000 training data.

## B    The comparison of $\sigma(\mathbf{T}\,\boldsymbol{\alpha} + \mathcal{K}\boldsymbol{\alpha})$ and $\mathbf{T}\,\boldsymbol{\alpha} + \sigma(\mathcal{K}\boldsymbol{\alpha})$

Figure 27 shows the comparison between GIT-Net with GIT mapping $\sigma(\mathbf{T}\,\boldsymbol{\alpha} + \mathcal{K}\boldsymbol{\alpha})$ and its variance in which the GIT mapping is replaced by $\mathbf{T}\,\boldsymbol{\alpha} + \sigma(\mathcal{K}\boldsymbol{\alpha})$. The tested hyper-parameters are $(C, K) = \{(2, 16), (4, 64), (8, 128), (16, 256), (32, 512)\}$.

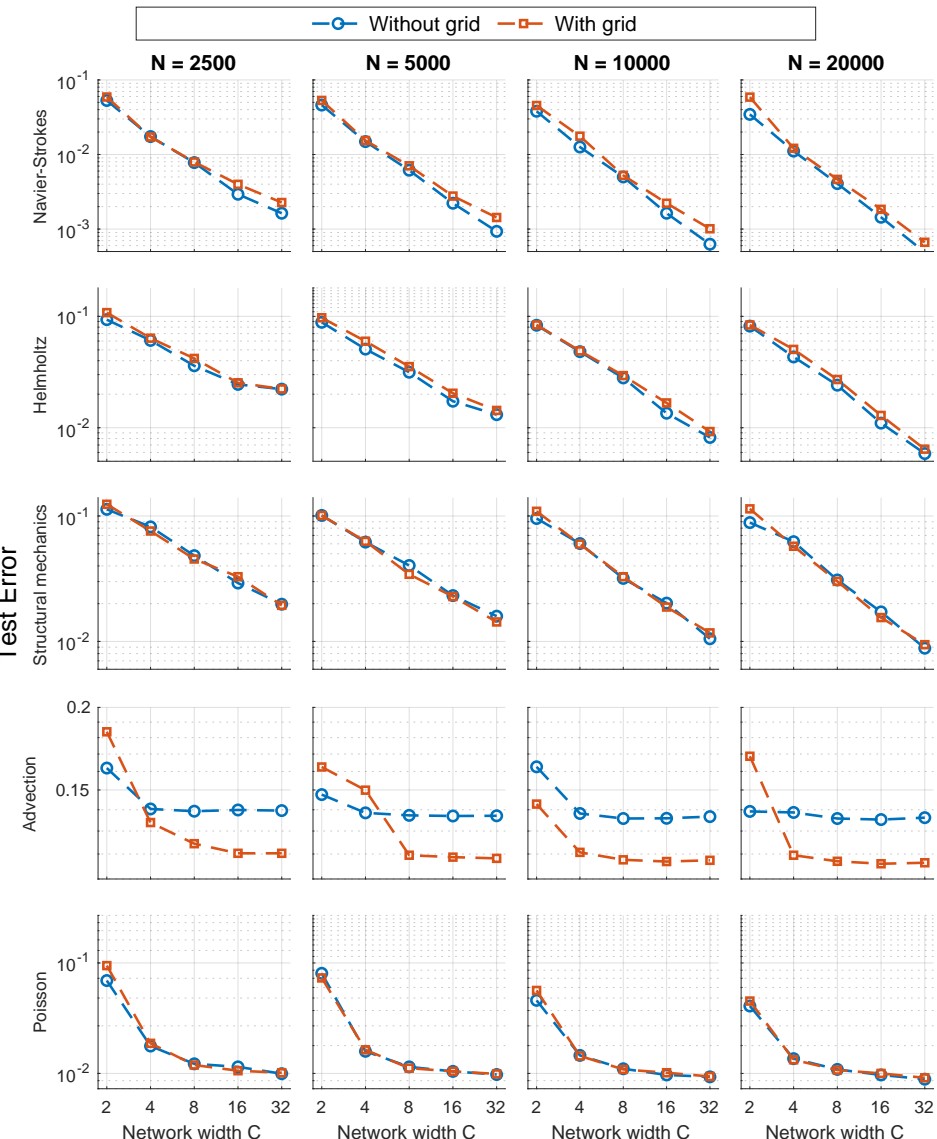

Figure 26: Test error of FNO method for the with-grid case and the without-grid case for all problems.

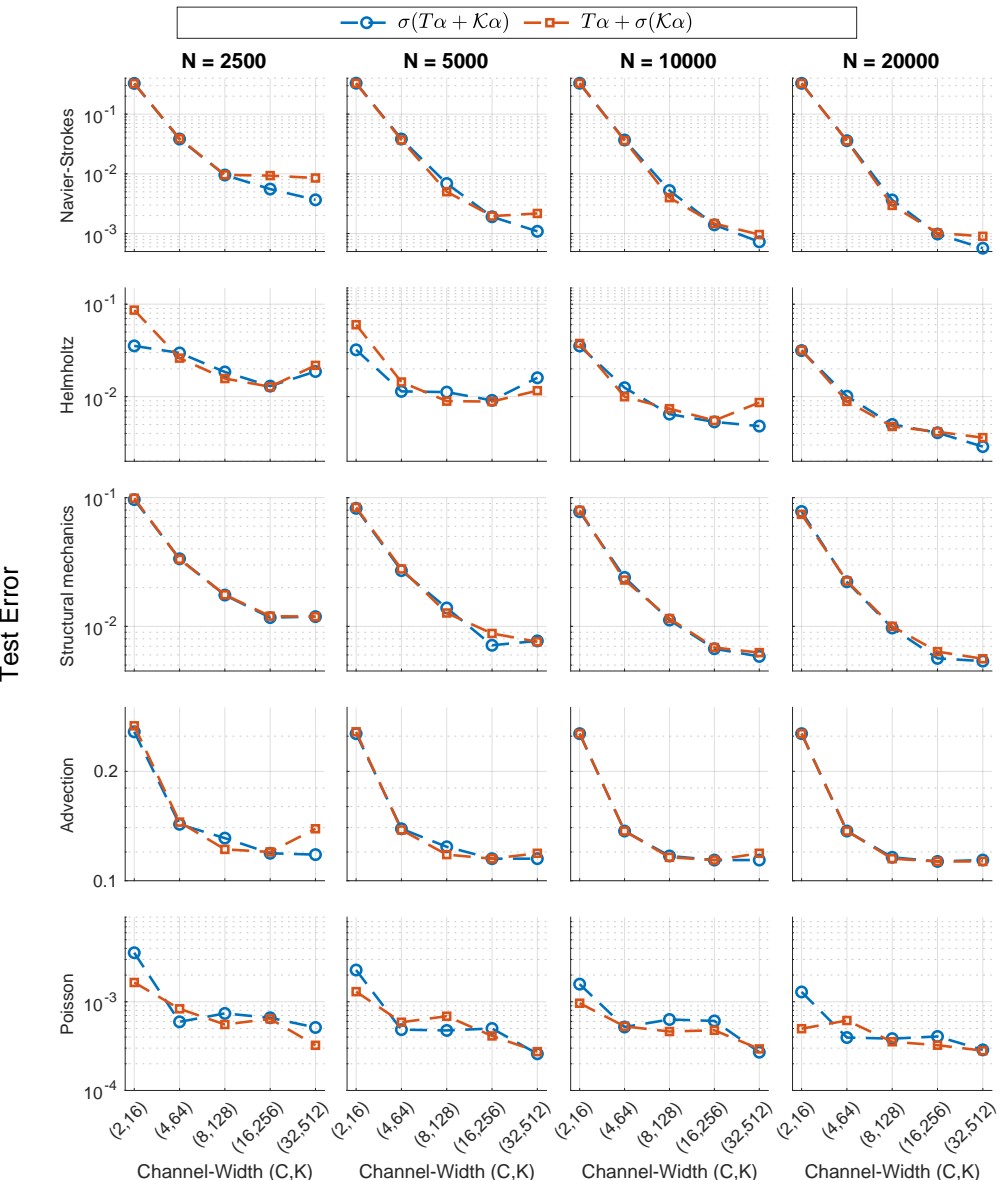

Figure 27: The comparison of test errors between GIT-Net and its variance.

