# OpenReview forum: "GIT-Net: Generalized Integral Transform for Operator Learning"
_TMLR — Accepted by TMLR_

### Review · Reviewer_DXV5 · 2023-02-03

**Summary Of Contributions:**

First, I'm not an expert in PDE operator learning, so my evaluation should be taken with a grain of salt. This paper proposes a neural architecture, GIT-Net, with residual blocks that use factorized linear transform inside. In some sense, GIT-Net compares to FNO, which is like MobileNet compares to ResNet. The ablation studies look promising. However, I also feel that the writing of this paper is language-heavy, and there is insufficient understanding provided to support the arguments.

My understanding of this work is that the authors propose a new neural network architecture, which is somewhat similar to MobileNet, and claim that the network achieve better empirical performance on PDE operator learning. No theoretical insights are provided and no understanding was provided at the moment. As I'm not an expert in PDE operator approximation, my review should be taken with a grain of salt. And I'm also not in a good position to evaluate whether the empirical advancement are convincing enough.



**Audience:**

Yes

**Broader Impact Concerns:**

I do not see any such concern.

**Claims And Evidence:**

No

**Requested Changes:**

In addition to my earlier comments, it is important to discuss what is to be learned from GIT-Net. My current take away is the following: GIT-Net is an efficient network architecture for doing PDE operator approximation. And it achieves better performance. This message is purely on the empirical side, so the authors should do a more thorough ablation study to show that GIT-Net can achieve better performance than all of the previous methods with a clear margin. And the code should be released to ensure that the results are reliable. And the paper should be significantly simplified.

If the authors wish to claim more on the theoretical insights, then much more works are needed to establish these arguments. I do not see any convincing insights in the current manuscript supported by convincing evidence.

**Strengths And Weaknesses:**

In the following, I only conclude a several concerns I have. I hope the authors can address them.

The notation used in this paper is "language-heavy." The overall takes are close to an image-to-image regression. The central proposal of the work is to use a block consisting of a skip connection and a factorized linear transform (pointwise linear transform and a depthwise linear transform, then pointwise linear transform again), followed by a GELU nonlinearity. The overall idea seems relatively straightforward. So I suggest the authors make an effort to simplify the notation. The language-heavy style makes such a simple idea look unnecessarily sophisticated.

The authors conclude that this method is better than FNO. However, insufficient understanding is provided. A MobileNet architecture would not necessarily perform better than a ResNet in terms of accuracy. The computation saving is expected since the architecture follows an efficient network design. The only surprising results are that some of the ablations seem to show that GIT-Net has less prediction error than FNO. But I think the authors need to discuss why. Otherwise, I would see this paper as parameter-tuning work.

On page 4, the first paragraph of Section 2.1, could the authors provide more evidence to support "This factorization allows one to mitigate overfitting issues significantly"?

On page 7,
, is the cdot a typo?

On page 11, equation (16), why not stick to the previous notation introduced in the first paragraph of Section 2.2?
 is confusing.

Section 5.2 - Error profiles are relatively uninformative; these only discuss what the error looks like. But why is the proposed architecture better? Why is this work not just parameter tunning? The overall setting of the work follows exactly FNO. So I think one can modify the neural architecture, tune the parameters to get better ablation results, and quickly write such a paper. Or many more papers like this. So please provide more insights.

On page 5, "The main assumption of the GIT-Net transform is that the different frequencies do not interact with each other." This statement seems quite misleading. If we think depthwise linear transform follows this assumption, then the two pointwise linear transforms do not, right? Maybe I misunderstood. But now, this statement looks like a forced one, which is trying to provide an unsupported theoretical insight into an engineering design choice.

So overall, I don't think this paper is ready for publication. Anyone with FNO code could modify the neural architecture with new network architectures and sufficient hyperparameter tuning to write such a paper. So the authors should provide in-depth insights and understanding to ensure the work is deeper than what I just mentioned.

Finally, I'm not an expert in this field. So if my understanding is significantly wrong, please ignore it, and I apologize in advance for the potentially naïve comments.

---

> ### Author Response · Authors · 2023-07-30
> **Response to Reviewer DXV5 (1)**
>
> **[Answers to the weakness]**
>
> **Q1** The notation used in this paper is "language-heavy." The overall takes are close to an image-to-image regression. ......
>
> **A:** We appreciate your careful review and apologize for the overly complex description of our network. To provide a clearer and more intuitive understanding, we have included a flow chart of the network, indicating the tensor sizes. Please see Figure 1 and Figure 2.
>
> Our network, GIT-Net, is inspired by integral transformations used to solve partial differential equations (PDEs). It follows a process of integral transform, algebraic equation solving, and integral inverse transform. The FNO method takes advantage of this capability.
>
> Unlike using a fixed Fourier basis (Fourier transform), our proposed GIT-Net explores a more general integral transform by employing  a pointwise linear transform in a deep network. The deepwise linear transform resembles the algebraic operations. Hence, the operator $\mathcal{K}$ within the network exhibits a resemblance to the sequential stages encompassing integral transform, algebraic operations, and integral inverse transform.
>
> We hope that the inclusion of the flow chart and this concise explanation clarifies our network architecture.
>
> **Q2** The authors conclude that this method is better than FNO. However, insufficient understanding is provided. A MobileNet architecture would not necessarily perform better than a ResNet in terms of accuracy. ......
>
> **A:** Our proposed GIT-Net presents a generalized architecture compared to FNO. Both architectures share a similar main block structure, denoted as $\sigma((\mathbf{T}+\mathcal{K})(\cdot))$, where $\mathbf{T}$ represents a linear transformation in the channel dimension, and $\mathcal{K}$ encompasses a combination of pointwise linear transformations and depthwise linear transformations, followed by another pointwise linear transformation.
>
> The key distinction lies in the nature of the pointwise linear transformation. While FNO employs the Fourier transform and its inverse as the pointwise linear transforms, GIT-Net employs two learnable linear transformations. This distinction allows GIT-Net to leverage the expressive power of deep neural networks and explore a broader range of function approximators beyond the limitations of the Fourier basis. Moreover, this structure allows GIT-Net to operate in an efficient manner, saving computation time, which cannot be done with a fixed Fourier transform.
>
> Consequently, GIT-Net exhibits improved performance compared to FNO in numerical simulations, as observed from a network architecture perspective. This finding supports the notion that the utilization of a more generalized function approximator in a wider parameter space contributes to the superior performance of GIT-Net over FNO. Additionally, it is noteworthy that GIT-Net exhibits significant computational time savings, especially when dealing with large-scale PDE problems.
>
> **Q3** On page 4, the first paragraph of Section 2.1, could the authors provide more evidence to support "This factorization allows one to mitigate overfitting issues significantly"?
>
> **A** Although the rationale behind this neural architecture is similar to the so-called "Depthwise Separable Convolutions" architecture [chollet2017xception], we agree with the reviewer that our set of numerical experiments does not directly provide evidence for this claim. For this reason, we have chosen to not include this remark. Thank you!
>
> **Q4** On page 7, , is the cdot a typo?
>
> **A** Yes, it is a typo and has been corrected.
>
> **Q5** On page 11, equation (16), why not stick to the previous notation introduced in the first paragraph of Section 2.2? is confusing.
>
> **A** We used the same notation of $\textbf{f}_i$ and $\textbf{g}_i$ in the first paragraph of Section 2.2 and Equation (16). In Section 2.2, we introduce $\mathcal{F}$ to denote PDE forward operator. While in Equation (16) we use $\mathsf{N}$ to denote predication $\mathsf{N}(\textbf{f})$. Therefore, the test error is defined by the distance between predication $\mathsf{N}(\textbf{f}_i)$ and the true output of PDE operator $\textbf{g}_i$.

---

> > ### Author Response · Authors · 2023-07-30
> > **Response to Reviewer DXV5 (2)**
> >
> > **Q6** Section 5.2 - Error profiles are relatively uninformative; these only discuss what the error looks like. But why is the proposed architecture better? Why is this work not just parameter tunning? ...... So please provide more insights.
> >
> > **A** Thank you for your detailed comments and suggestions.
> >
> > Regarding the feedback on the error profiles, we acknowledge that they may not fully convey the comparative advantages of the proposed GIT-Net architecture. To provide more insights, we have included additional visualizations in the updated manuscript, specifically histograms of test errors, which offer a clearer representation of the error distribution and comparisons between neural operators.
> >
> > In response to the reviewers' comments, we appreciate the opportunity to address the concerns and provide further insights into our work on deep learning with neural operators.
> >
> > The GIT-Net architecture draws inspiration from FNO, which effectively employs the Fourier transform for solving differential equations. FNO utilizes a specific operator $\mathcal{K}$ parameterized by a neural network to handle the Fourier transform. While FNO has been successful, it is limited in its applicability to certain grids and specific types of PDEs. Moreover, for some PDEs, the Fourier transform may not even exist, restricting its usage in those scenarios.
> >
> > To overcome these limitations and enhance the versatility of the approach, GIT-Net introduces a more general integral transform, also parameterized by a neural network within the operator $\mathcal{K}$. This alternative integral transform, as described in Equation (5) of the paper, offers a broader range of linear transformations compared to the Fourier transform. By employing this more general integral transform, GIT-Net is capable of exploring a wider and more diverse approximator of function spaces, effectively addressing the limitations of FNO and extending the potential applications of neural operators.
> >
> > Moreover, GIT-Net exhibits superior performance in PDE problems with complex computational regions, attributed to two key factors. Firstly, its utilization of a more general approximate function approximator allows for greater flexibility beyond FNO's capabilities. Secondly, GIT-Net avoids the need for function interpolation, a requirement in FNO within complex regions, thereby enhancing its predictive capabilities.
> >
> > In summary, the distinctive characteristics of GIT-Net, including its expanded search for approximation space and reduced computation through PCA, cannot be solely achieved through parameter tuning. These features solidify GIT-Net as a superior alternative to FNO for neural operators and differential equation solving.
> >
> > **Q7** On page 5, "The main assumption of the GIT-Net transform is that the different frequencies do not interact with each other." This statement seems quite misleading. If we think depthwise linear transform follows this assumption, then the two pointwise linear transforms do not, right?......
> >
> > **A** The assumption of non-interference between frequencies in GIT-Net's transformation is based on the integral transformation method for solving PDEs. This assumption signifies that computations occur within channels and not between different frequencies. For example, in Fourier transform, $F(\omega)$ denotes the Fourier transform of the function $f$, and algebraic operations are performed between frequency domain functions, such as $F(\omega)$ and $H(\omega)$, without involving operations between distinct frequency components like $F(\omega_1)$ and $F(\omega_2)$. Here, the convolution operation with a kernel size greater than 1 and the linear operation does not satisfy the assumption. We made this assumption to align with the principles of integral transform used in PDE solving.
> >
> > **Q8** So overall, I don't think this paper is ready for publication. Anyone with FNO code could modify the neural architecture with new network architectures and sufficient hyperparameter tuning to write such a paper. So the authors should provide in-depth insights and understanding to ensure the work is deeper than what I just mentioned.
> >
> > **A** Our proposed GIT-Net goes beyond simple parameter tuning. The primary strength of GIT-Net lies in its ability to explore a more general function approximator compared to FNO. This fundamental distinction enables GIT-Net to achieve superior prediction accuracy. Additionally, the implementation of model reduction techniques significantly reduces the forward operation time of the network from $\mathcal{O}(N \log N)$ to $\mathcal{O}(N)$, which is particularly beneficial for large-scale PDE problems. Our numerical experiments confirm the efficacy of these enhancements.
> >
> > By incorporating these advancements, our work provides valuable insights and understanding that extend beyond the scope of mere parameter modifications. We believe that these contributions justify the readiness of our paper for publication.

---

> > > ### Author Response · Authors · 2023-07-30
> > > **Response to Reviewer DXV5 (3)**
> > >
> > > **[Answers to the requested changes]**
> > >
> > > **Q** In addition to my earlier comments, it is important to discuss what is to be learned from GIT-Net. My current take away is the following: GIT-Net is an efficient network architecture for doing PDE operator approximation. And it achieves better performance. This message is purely on the empirical side, so the authors should do a more thorough ablation study to show that GIT-Net can achieve better performance than all of the previous methods with a clear margin. And the code should be released to ensure that the results are reliable. And the paper should be significantly simplified.
> > >
> > > If the authors wish to claim more on the theoretical insights, then much more works are needed to establish these arguments. I do not see any convincing insights in the current manuscript supported by convincing evidence.
> > >
> > > **A** Our code has been released with the first version of the submitted manuscript and is noted in the footnotes. Anyone can download (Github: https://anonymous.4open.science/r/Operator-Learning-Generalized-integral-transform-neural-network-A2B8) and test it.
> > >
> > > Our GIT-Net is inspired by FNO that learns a kernel integral operator parameterized by a neural network. Our proposed GIT-Net effectively addresses certain limitations of FNO. These limitations include the necessity for interpolation in handling complex regions and accommodating differential dimensions between input and output, the computationally expensive Fourier transform for addressing large-scale problems, as well as the restricted applicability to a limited range of function types. Through the utilization of our GIT-Net, we facilitate the acquisition of an analogous structure for integral transforms. This, in turn, enables a broader exploration of function approximation as opposed to being confined to fixed approximators such as the Fourier basis in FNO. The application of PCA reduction not only facilitates mesh-free and efficient implementation but also enhances the capability to handle complex and varying dimensions.
> > >
> > > Our numerical simulations demonstrate that the advantages offered by GIT-Net cannot be achieved solely through parameter tuning of its predecessors, FNO and PCA-Net, from which GIT-Net originates.

---

### Review · Reviewer_gw94 · 2023-03-04

**Summary Of Contributions:**

This paper proposed a new neural architecture called GIT-net. GIT-net is an operator learner that learns the PDE solution map under the standard end2end pipeline. One key ingredient is to treat both the channel/latent dimension and the discretization dimension as the coefficients of a "basis" (from PCA). The other draws inspiration from the depthwise separable convolution to create a learnable combination of different bases. Evaluation results are presented along with some other baselines.

**Audience:**

Yes

**Claims And Evidence:**

No

**Requested Changes:**

### Requested changes
- It would be nice to add a diagram marking the tensors' dimensions in Step 1-4 in Section 2.1.

- It would be nice to see a comparison of (9) with $T\boldsymbol{\alpha} + \sigma(\mathcal{K}\boldsymbol{\alpha})$.

- It is worthwhile to mention explicitly that $\{b_k\}$ comes from PCA's $V$ matrix (assuming the data has shape `(N_data, N_discretization)`) on page 6 as well as in the beginning of Section 2.1 (instead of saying just "PCA bases"), since there are two "bases" in PCA. Another thing is that, as a reader, I initially found the basis analogy in Section 2.1 somewhat puzzling until reaching the PCA part.

- In all examples for GIT, `np.linalg.svd` is performed, and the spectral information of the data bulk is furthermore used to determine the number of bases (a hyper-parameter of the model). I understand that FNO does not need SVD to change basis, but I think this is somewhat unfair to FNO.

- The evaluation cost on page 18 is somewhat unfair to FNO as PCA is involved in model reduction.

- In FNO's code, for example, `FNO2d` module in the `model_ol.py` file, the lines ` grid = self.get_grid(x.shape, x.device)` and `x = torch.cat((x, grid), dim=-1)` have been commented out. Is FNO's benchmarks tested under this set-up??? If so, I think this is unfair for FNO. I understand that it is nontrivial to adapt this step for the triangular domain case, but for the square domain case, these lines should not be commented out.

### Minor things that need to be clarified or fixed
- When presenting the core architecture on page 5, sometimes bold-faced checked ${\boldsymbol{\alpha}}$ is used, while non-boldfaced is used as well, for example, "$\check{\alpha}\in \mathbb{R}^{C,K}$".
- Page 4: "Assume that each coordinate of the input function", the "each coordinate" part is confusing. It should be "each coordinate of the image of the input function" or something similar.
- Page 5: Section 2.1, bullet 2, the boldfaced $\boldsymbol{\alpha}$ is used, but it needs to be formally defined in bullet 1.
- Page 5: Section 2.1, the author uses the term "change of coordinate" a lot, but the coordinate is not changed (every basis uses $\boldsymbol{x}$ as coordinates). I think here the author meant to say "change of basis".
- Page 5: After PCA, the neural network has to scale the input functions in the basis dimension (`d_width` in the `lift_d` part). Is this approach still make this neural network achieve invariance with regard to the discretization size?
- Page 5: $\mathcal{K}$ here is used as a linear operator, yet in the example section, it is used as the notation for the RBF kernel.
- Page 6: the term "functional mapping" is confusing as functionals map a function to a scalar; please use "operator".
- Page 10: if the permeability is a constant, then this is not modeling a porous medium. The BCs are not modeling in/out-flows either, thus the equation is simply a Poisson equation, not Darcy.
- Figure 8-12: Can another set of figures be added maybe in the appendix such that the inputs to all models be the same? Another nitpicking is that, in the caption of the figures, it says "smallest test error", but later it says "median test error", a more precise description should be used.
- Page 18: "FNO method shows severe oscillations, which is likely due to interpolation error". What is the "interpolation error" referred to here?
- Page 20: I think "favorable properties for learning PDE operators" should be "learning forward PDE operators" as no inverse problem example is presented. The paper does mention the loss function for the Bayesian inverse problem on page 3, and I suggest deleting this presentation on page 3.
- Missing references about designing neural networks for PDE operator learning:
    - Gupta et al. Multiwavelet-based operator learning for differential equations. NeurIPS 2021.
    - Cao. Choose a transformer: Fourier or galerkin. NeurIPS 2021.
    - Kissas et al. Learning Operators with Coupled Attention. JMLR 2022.
    - Ong et al. Integral autoencoder network for discretization-invariant learning. JMLR 2022.
    - Brandstetter et al. Message passing neural PDE solvers. ICLR 2022.
    - Huang et al. Meta-auto-decoder for solving parametric partial differential equations is in NeurIPS 2022.
- On several occasions, "i.e" or "i.e." should be "i.e.,".
- Page 20: "ongoing work" -> "an ongoing work".

**Strengths And Weaknesses:**

### Strengths
- The paper conducts a comprehensive comparison in several forward PDE operator learning problems.
- The architectural connection of the depthwise separable convolution and the combination of basis is interesting.
- The application of PCA enables the model to handle problems on non-tensor-product-type grids nicely.

### Weaknesses
- The GIT stands for "generalized ***integral*** transform", but where is the ***integral***? There is no component in the neural network computing an integral transform. The analogy with a learnable integral transform is not clear at all.
- PCA is used for the whole bulk data, i.e., a full SVD is done for a matrix of size `(N_data, N_discretization)`. The full SVD is very memory hungry! This defeats the purpose of introducing deep learning (using mini-batch to avoid operations in bulk, see also the requested changes below).
- The extension to PCA-Net is somewhat limited theoretically.
- The details in writing could use some proofreading and the presentation needs polishing.

---

> ### Author Response · Authors · 2023-07-30
> **Response to Reviewer gw94 (1)**
>
> **[Answers to the weakness]**
>
> **Q1** The GIT stands for "generalized integral transform", but where is the integral? There is no component in the neural network computing an integral transform. The analogy with a learnable integral transform is not clear at all.
>
> **A:** In our paper, we propose GIT-Net as a novel architecture inspired by the success of FNO (Fourier Neural Operator), which utilizes the Fourier transform to solve differential equations. However, we acknowledge that not all differential equations can be effectively solved using a single integral transform, such as the Fourier transform, especially for unstable systems.
>
> The main motivation behind GIT-Net is to introduce a learnable integral transform that can adapt to a wide range of differential equations. Our approach is not limited to a specific integral transform, such as the Fourier transform, but rather explores the concept of a learned integral transform that is suitable for different scenarios.
>
> To clarify the analogy with integral transforms, we present the linear expansions of $f^c(\textbf{x})$ under two sets of basis functions, denoted by $[b_k]\_{k\geq 1}$ and $[\check{b}\_k]\_{k\geq 1}$ in Equations (4) and (5) respectively. While the mapping $f^c\mapsto [\check{\alpha}\_{c,k}]$ becomes a Fourier transform when $[\check{b}\_k]$ represents the Fourier basis, we generalize this concept to encompass any basis functions. In this case, the mapping $\alpha\_{c,k}\rightarrow \check{\alpha}\_{c,k}$ is a learned linear transform, making $\check{\alpha}\_{c,k}$ analogous to a spectrum, and the mapping $f^c\mapsto [\check{\alpha}\_{c,k}]$ serves as an analogous integral transform.
>
> In Equation (6), $\check{ \beta }\_{c,k} = \sum\_{d=1}^C \mathbf{D}\_{d,c,k}\, \check{ \alpha }\_{d,k}$ represents the operator in frequency space, which aligns with the integral transform analogy.
>
> Overall, Step 2-Step 4 in Section 2.1 can be analogous to forward integral transform, operation in frequency space, and inverse integral transform.
>
> We understand that the concept of spectrum for $\check{\alpha}\_{c,k}$ may not strictly adhere to mathematical definitions, but our primary objective is to design GIT-Net as a versatile neural network that resembles integral transforms, such as the Fourier and Laplace transforms.
>
> In summary, GIT-Net aims to explore a formula in the form of a neural network that effectively performs integral transform-like operations suitable for a wide range of differential equations. We believe this approach opens up new possibilities for solving complex problems in various domains. Thank you for providing us with this opportunity to clarify our work.
>
> **Q2** PCA is used for the whole bulk data, i.e., a full SVD is done for a matrix of size (N_data, N_discretization). The full SVD is very memory hungry! This defeats the purpose of introducing deep learning (using mini-batch to avoid operations in bulk, see also the requested changes below).
>
> **A:** A full SVD is not necessary. We can use library python `numpy.linalg.svd` with parameter `full_matrices=False` to implement SVD.  Even N_data and N_discretization are big, only a small amount of basis functions are needed and stored. Once the basis functions are found and stored, only the coefficients are implemented in the network using a mini-batch. The computation in the network for original data is **(batch_size, N_discretization)**, and for GIT is **(batch_size, N_basis)**. For example, in the two-dimensional Navier-Stokes problem, **N_discretization = 64*64=4096**. After using PCA, the length of coefficients is **N_basis = 200**. Therefore, the size of the input of the deep neural network is reduced by about **(4096-200)/4096=95%**. That's how PCA makes GIT-Net efficient.
>
> While we acknowledge that PCA may incur some offline computation time on SVD, this preprocessing step significantly speeds up the online computation when predicting solutions of equations repeatedly. This aligns well with the iterative nature of neural operator-based methods.

---

> > ### Author Response · Authors · 2023-07-30
> > **Response to Reviewer gw94 (2)**
> >
> > **Q3** The extension to PCA-Net is somewhat limited theoretically.
> >
> > **A:** Although our GIT-Net and PCA-Net both use PCA reduction, they differ in their intermediate representations and transformations.
> >
> > Our GIT-Net's novelty lies in the operator $\mathcal{K}$, which is inspired by the success of integral transform in PDE solving. The operator $\mathcal{K}$ can be an analogy to PDE solvers based on integral transform. For example, FNO is a special case of using the Fourier transform as an integral transform. Our GIT-Net allows a wider search than any fixed transform.
> >
> > Compared to the single channel of PCA-Net that only contains pointwise linear transformation, our GIT-Net has a multi-channel two-dimensional matrix representation that also involves depthwise linear transformation. However, when $C=1$, depthwise linear transformation in $\mathcal{K}$ of GIT-Net reduces to a scalar product, resulting GIT-Net only containing pointwise linear transformations and is equivalent to PCA-Net. When $C>1$, GIT-Net’s deeper structure and the neural network parameterized $K$ result in superior fitting ability compared to PCA-Net. Our simulations also demonstrate that simply increasing PCA-Net’s size to match GIT-Net’s intermediate representation does not yield similar predictive performance.
> >
> > **Q4** The details in writing could use some proofreading and the presentation needs polishing.
> >
> > **A:** Writing is improved and the paper is polished.
> >
> > **[Answers to the requested changes]**
> >
> > **Q5** It would be nice to add a diagram marking the tensors' dimensions in Steps 1-4 in Section 2.1.
> >
> > **A:** Thanks for your advice. We already added Figure 1 and Figure 2 in the updated version as you suggested.
> >
> > **Q6** It would be nice to see a comparison of (9) with $T\boldsymbol{\alpha} + \sigma (\mathcal{K}\boldsymbol{\alpha})$.
> >
> > **A:** Thank you for your valuable feedback. We have conducted a thorough comparison between $\sigma(T\boldsymbol{\alpha} + \mathcal{K}\boldsymbol{\alpha})$ and $T\boldsymbol{\alpha} + \sigma (\mathcal{K}\boldsymbol{\alpha})$ by implementing all the PDE problems mentioned in the paper. The test errors were plotted in Figure 26 in Appendix B of the updated manuscript.
> >
> > Specifically, we focused on the test error of the Navier-Stokes equation and Helmholtz equation, using 2500 training data points. The results, as presented in the following two tables, demonstrate that $\sigma(T\boldsymbol{\alpha} + \mathcal{K}\boldsymbol{\alpha})$ achieved significantly smaller test errors compared to the alternative approach of $T\boldsymbol{\alpha} + \sigma (\mathcal{K}\boldsymbol{\alpha})$.
> >
> > Moreover, we extended the comparison to include simulations for all PDE problems as depicted in Figure 26. The outcomes indicated that both architectures performed similarly in these tests.
> >
> > *Table 1: Test error of Navier-Stokes equation on 2500 testing data*
> >
> > |                            (C, K)                            | (2, 16) | (4, 64) | (8, 128) | (16, 256) | (32, 512) |
> > | :----------------------------------------------------------: | :-----: | :-----: | :------: | :-------: | :-------: |
> > | $T\boldsymbol{\alpha} + \sigma (\mathcal{K}\boldsymbol{\alpha})$ |  32.6%  |  3.89%  |  0.961%  |  0.934%   |  0.852%   |
> > | $\sigma(T\boldsymbol{\alpha} + \mathcal{K}\boldsymbol{\alpha})$ |  32.7%  |  3.83%  |  0.955%  |  0.556%   |  0.367%   |
> >
> > *Table 2: Test error of Helmholtz equation on 2500 testing data*
> >
> > |                            (C, K)                            | (2, 16) | (4, 64) | (8, 128) | (16, 256) | (32, 512) |
> > | :----------------------------------------------------------: | :-----: | :-----: | :------: | :-------: | :-------: |
> > | $T\boldsymbol{\alpha} + \sigma (\mathcal{K}\boldsymbol{\alpha})$ |  8.63%  |  2.61%  |  1.57%   |   1.28%   |   2.18%   |
> > | $\sigma(T\boldsymbol{\alpha} + \mathcal{K}\boldsymbol{\alpha})$ |  3.54%  |  2.97%  |  1.85%   |   1.30%   |   1.87%   |
> >
> > **Q7** It is worthwhile to mention explicitly that $b_k$ comes from PCA's matrix (assuming the data has shape (N_data, N_discretization)) on page 6 as well as in the beginning of Section 2.1 (instead of saying just "PCA bases"), since there are two "bases" in PCA. Another thing is that, as a reader, I initially found the basis analogy in Section 2.1 somewhat puzzling until reaching the PCA part.
> >
> > **A:** Thanks for the comments and sorry for the confusion we caused. We added the explanations of PCA as you suggested.

---

> > > ### Author Response · Authors · 2023-07-30
> > > **Response to Reviewer gw94 (3)**
> > >
> > > **Q8** In all examples for GIT, `np.linalg.svd` is performed, and the spectral information of the data bulk is furthermore used to determine the number of bases (a hyper-parameter of the model). I understand that FNO does not need SVD to change basis, but I think this is somewhat unfair to FNO.
> > >
> > > **A:** Both the SVD of GIT-Net and the Fourier transform of FNO can be regarded as the calculation of spectral information. The difference is that GIT only does it offline and then uses the spectral information to train the network, but FNO does it online. Therefore, it's a trade-off between saving online computation and saving offline computation. For the application that needs to solve a class of PDEs repeatedly, GIT-Net does not need SVD after training is finished if PCA bases are stored while training, so GIT-Net is more advantageous in this scenario. And the computation of GIT-Net for each prediction is $\mathcal{O}(N)$ when assuming the dimension of discretization of input functions $N$. However, the computation of FNO is $\mathcal{O}(N\log(N))$ for each prediction.
> > >
> > > Indeed, in all our simulations involving various large-scale PDE problems such as Navier-Stokes equation, Helmholtz equation, structural mechanics problems, and Poisson equation, GIT-Net consistently exhibits significantly reduced training time compared to FNO, given the same training data. However, it is important to note that for small-scale PDE time, specifically for the advection equation, GIT-Net requires more training time than FNO.
> > >
> > > **Q9** The evaluation cost on page 18 is somewhat unfair to FNO as PCA is involved in model reduction.
> > >
> > > **A:** As we explained before, there is a trade-off between saving online computation and saving offline computation. As we analyze, GIT-Net is more advantageous in the scenario that needs saving online computation. This evaluation cost on Page 18 is to illustrate the advantage of model reduction when iteratively solving forward problems.
> > >
> > > Because the neural operators discussed in our paper are all applied to scenarios that need to repeatedly solve a type of PDEs, such as iteratively solving inverse problems, uncertain quantification, and other applications that need to repeatedly solve forward problems. Therefore, what we mainly care about is whether the calculation time can be reduced when using the trained network operator to solve the forward problem, that is, the evaluation cost.
> > >
> > > Theoretically, when PCA is used for network reduction, although the amount of offline calculations increases, it can effectively reduce the number of online calculations for evaluation, which is consistent with our expectations for neural operators in this usage scenario. Although FNO has almost no offline calculation, it has a large amount of online calculation. Therefore, when repeatedly solving the forward problems in the application, FNO requires a larger amount of calculation than the network operator using PCA.
> > >
> > > Indeed, combining online and offline computational time, in our simulations, GIT-Net costs less training and testing time for large-scale PDE problems including Navier-Stokes equation, Helmholtz equation, structural mechanics problems, and Poisson equation. FNO costs less training and testing time for small-scale PDE problem advection equations.
> > >
> > > **Q10** In FNO's code, `FNO2d` in `model_ol.py` file, the lines `grid = self.get_grid(x.shape, x.device)` have been commented out. Is FNO's benchmarks tested under this set-up???
> > >
> > > **A:** As you suggested we considered the grid coordinates as part of input and compared the prediction results with those without grid coordinates for PDE problems in our paper. Please see Appendix A in the updated manuscript. It is found that the two results are similar except for the advection equation. For the advection equation, the with-grid case got a smaller test error for most hyperparameters. The FNO results in the main part of the manuscript are replaced by the with-grid case (for the triangular grid, the interpolated rectangular grids are used).
> > >
> > >
> > > **[Minor things that need to be clarified or fixed]**
> > >
> > > - When presenting the core architecture on page 5, sometimes bold-faced checked $\boldsymbol{\alpha}$is used, while non-boldfaced is used as well, for example, "$\check{\alpha}\in \mathbb{R}^{C,K}$".
> > >
> > >   **A:** It was corrected to $\check{\boldsymbol\alpha}\in \mathbb{R}^{C,K}$.
> > >
> > > - Page 4: "Assume that each coordinate of the input function", the "each coordinate" part is confusing. It should be "each coordinate of the image of the input function" or something similar.
> > >
> > >   **A:** We would like to use "each component". It is assumed that there are multiple input functions that may contain one or several boundary conditions, initial functions, and force terms. Then the input functions can be assumed a vector-valued function $\boldsymbol{f}=(f^1, f^2, \dots, f^C)$. In the language of deep learning, $\boldsymbol{f}$ is a multiple-channel variable and its length $C$ is the number of channels.

---

> > > > ### Author Response · Authors · 2023-07-30
> > > > **Response to Reviewer gw94 (4)**
> > > >
> > > > - Page 5: Section 2.1, bullet 2, the boldfaced $\boldsymbol{\alpha}$ is used, but it needs to be formally defined in bullet 1.
> > > >
> > > >   **A:** Define $\boldsymbol{\alpha} = [\alpha_{c,k}]\in \mathbb{R}^{C,K}$ .
> > > >
> > > > - Page 5: Section 2.1, the author uses the term "change of coordinate" a lot, but the coordinate is not changed (every basis uses � as coordinates). I think here the author meant to say "change of basis".
> > > >
> > > >   **A:** Yes. It means "change of basis". We have already corrected them as you suggested.
> > > >
> > > > - Page 5: After PCA, the neural network has to scale the input functions in the basis dimension (`d_width` in the `lift_d` part). Is this approach still make this neural network achieve invariance with regard to the discretization size?
> > > >
> > > >   **A:** Yes. When discretization size is changed, the discretization of basis function $b_k(x)$ is changed by downsampling or upsampling, but the coefficient $\boldsymbol{\alpha}$ still makes the neural network work. According to the numerical simulations in the paper "Model Reduction And Neural Networks For Parametric PDEs (*Kaushik Bhattacharya, Bamdad Hosseini, Nikola B. Kovachki, and Andrew M. Stuart*)", if the network is trained in coarser discretization, the test error will become larger. In contrast, if the network is trained with finer discretization, the test error will become the same or slightly smaller. It is a general phenomenon for discretization-invariance network architecture. Anyway, once the network is trained in a specified discretization, it still works for different discretization but got a better prediction for coarser discretization, and a worse prediction for finer discretization.
> > > >
> > > > - Page 5: $\mathcal{K}$ here is used as a linear operator, yet in the example section, it is used as the notation for the RBF kernel.
> > > >
> > > >   **A:** We use normal symbol $\mathcal{R}$ to notate RBF kernel.
> > > >
> > > > - Page 6: the term "functional mapping" is confusing as functionals map a function to a scalar; please use "operator".
> > > >
> > > >   **A:** Thanks for your valuable feedback. It was corrected as you suggested.
> > > >
> > > > - Page 10: if the permeability is a constant, then this is not modeling a porous medium. The BCs are not modeling in/out-flows either, thus the equation is simply a Poisson equation, not Darcy.
> > > >
> > > >   **A:** Thanks for your valuable feedback. All "Darcy" has been corrected to "Poisson".
> > > >
> > > > - Figure 8-12: Can another set of figures be added maybe in the appendix such that the inputs to all models be the same? Another nitpicking is that, in the caption of the figures, it says "smallest test error", but later it says "median test error", a more precise description should be used.
> > > >
> > > >   **A:** Sorry for the confusion caused. We added examples from the same samples for all neural network operators.
> > > >
> > > >   In the caption of the figures you mentioned, the first "smallest test error" means that we choose the hyperparameters that achieve the smallest test error (average over all test data). The second "median test error" means that we choose the prediction that achieves median test error in all test data. The third "largest test error" means that we choose the prediction that achieves the largest test error in all test data. For different problems, the median-error case and worst-error case may be different. In the updated manuscript, we explained how we choose these figures to show the distribution of test errors on 20000 test data of the best hyperparameters for each neural operator and each PDE problem. To furthermore provide statistical information on the test errors, we show the histogram of the test errors in Figure 11, 14, 17, 20 and 23.
> > > >
> > > > - Page 18: "FNO method shows severe oscillations, which is likely due to interpolation error". What is the "interpolation error" referred to here?
> > > >
> > > >   **A:** When solving the problem with nonuniform discretization using FNO method, we need to interpolate the input and output functions in uniform Cartesian grids to make Fourier transform work. According to different resolution and interpolation methods, there exists an interpolation error.
> > > >
> > > > - Page 20: I think "favorable properties for learning PDE operators" should be "learning forward PDE operators" as no inverse problem example is presented. The paper does mention the loss function for the Bayesian inverse problem on page 3, and I suggest deleting this presentation on page 3.
> > > >
> > > >   **A:** It was deleted as you suggested.
> > > >
> > > > - Missing references about designing neural networks for PDE operator learning:
> > > >
> > > >   **A:** Thanks for your comments. These references have been added.
> > > >
> > > > - On several occasions, "i.e" or "i.e." should be "i.e.,".
> > > >
> > > >   **A:** They were corrected as you suggested.
> > > >
> > > > - Page 20: "ongoing work" -> "an ongoing work".
> > > >
> > > >   **A:** It was corrected as you suggested.

---

> > > > > ### Comment · Reviewer_gw94 · 2023-08-17
> > > > > **Some further comments from Reviewer gw94**
> > > > >
> > > > > Sorry for the delayed response. I think with the revision and additional ablation study the paper is much better and a decent submission to TMLR. Here are some further comments.
> > > > > - The biggest complaint from me is still that I cannot get over with the name "generalized integral transform". The author said eq. (6) "represents the operator in frequency space", note this is only true when the data has a spectral decay, so that when expanding in the basis obtained from SVD, the variance happens to align with the frequency (higher-frequency components happen to be the ones with the lowest variance).
> > > > > - Personally, I would name the model "generalized orthogonal transform". Because the expansion in SVD bases does preserve inner product: $\langle f, g\rangle = \langle [\alpha_f], [\alpha_g]\rangle$ (with the correct normalization using singular values). Yet I leave this to the authors to decide.
> > > > > - If `full_matrices=False` is used, then some literature of rank-$k$ SVD or incremental SVD should be cited. I also suggest adding one more reference closely related:
> > > > >     - Zijie Li et al. Transformer for Partial Differential Equations’ Operator Learning. *TMLR*.
> > > > > - A minor nitpicking: "Karhunen–Loeve" -> "Karhunen–Lo&#233;ve" it should be `\'{e}`.
> > > > > - By the way, I would not call FNO "has a large amount of online calculation" due to the FFT (the fastest way to transform the functions to orthogonal bases, probably human can ever create) being applied online, or treat the online FFT/iFFT as a significant overhead. Lots of iterative solvers such as Krylov subspace-based method use FFT as a preconditioner for a reason.

---

> > > > > > ### Author Response · Authors · 2023-09-26
> > > > > > **Response to Reviewer gw94**
> > > > > >
> > > > > > **1** The biggest complaint from me is still that I cannot get over with the name "generalized integral transform". ... ...
> > > > > >
> > > > > > **A** Thank you for your thoughtful feedback. We do agree that the effectiveness of the SVD expansion relies on the spectral decay of the data. Firstly, our data and basis conform to this condition due to the assumed inherent smoothness characteristics. For example, the input functions, derived from Gaussian random fields, exhibit this smoothness. Furthermore, our basis, generated through reduced SVD, maintains a certain approximation accuracy for the training data, leading to Equation (4). Secondly, while Equation (6) invokes an analogy to Fourier basis and $\check{\beta}_{c,k}$ is akin to frequency, it does not precisely represent mathematical frequencies.
> > > > > >
> > > > > > The choice of 'generalized integral transform' for naming our neural network is motivated by its inspiration from integral transforms used in the solution of partial differential equations (PDEs). Our approach follows a process akin to integral transform, algebraic equation solving, and integral inverse transform, mirroring the structure found in Equations (5), (6), and (7) respectively. Notably, the FNO method effectively leverages this capability. Our GIT-Net shares a similar architecture with FNO, and this is why we draw comparisons between GIT-Net and FNO. Through the application of GIT-Net, we aim to foster an analogous framework for integral transforms. In short, the name "generalized integral transform" is an acknowledgement that the proposed method is inspired from the FNO method.
> > > > > >
> > > > > > ---
> > > > > >
> > > > > > **2** Personally, I would name the model "generalized orthogonal transform". Because the expansion in SVD bases does preserve inner product: $\left<f, g\right> = \left<[\alpha_f], [\alpha_g]\right>$ ... ...
> > > > > >
> > > > > >  **A** Thank you for your valuable feedback. We appreciate your concern regarding the name “generalized integral transform“. Allow us to provide further clarification. Our choice of naming the network ”generalized integral transform“ extends beyond a mere label; it underlines our intent to construct a neural network architecture that draws inspiration from the mathematical technique known as the "integral transform". This mathematical tool is used in various fields of science and engineering to solve a wide range of problems, particularly those involving differential equations. Prominent integral transforms include the Fourier transform and the Laplace transform.
> > > > > >
> > > > > >  The fundamental idea behind integral transforms is to convert a problem from its original domain (typically time or space) into a new domain (often frequency or Laplace space) where the problem becomes simpler to solve. Once the solution is obtained in this transformed domain, it can often be inverted to obtain the solution in the original domain. Equations (5), (6), and (7) are analogies to this process. However, our network does not strictly correspond to each step of the mathematical procedure for solving PDEs through integral transform. Rather, we utilize the term "generalized" to signify that we employ a neural network to capture a structurally similar framework, although not a precise mathematical correspondence.
> > > > > >
> > > > > >  ---
> > > > > >
> > > > > > **3** If ```full_matrices=False``` is used, then some literature of rank-SVD or incremental SVD should be cited.
> > > > > >
> > > > > > **A** Thank you, it is an important remark. We have now added a small section on scalable approaches for computing the first few dominant Principal Components (PCs). We advocate leveraging modern methods based on randomized numerical linear algebra.
> > > > > >
> > > > > >
> > > > > > ---
> > > > > >
> > > > > > **4** I also suggest adding one more reference closely related:
> > > > > > Zijie Li et al. Transformer for Partial Differential Equations’ Operator Learning. TMLR.
> > > > > >
> > > > > > **A** Thank you for your suggestion -- this text is indeed related. It is now added in the section when transformer architectures and attention mechanisms are mentioned.
> > > > > >
> > > > > >
> > > > > > ---
> > > > > >
> > > > > >
> > > > > > **5** A minor nitpicking: "Karhunen–Loeve" -> "Karhunen–Loéve" it should be \'{e}.
> > > > > >
> > > > > > **A** Thank you. It is now correct.
> > > > > >
> > > > > >
> > > > > > ---
> > > > > >
> > > > > >
> > > > > > **6** By the way, I would not call FNO "has a large amount of online calculation" due to the FFT (the fastest way to transform the functions to orthogonal bases, probably human can ever create) being applied online, or treat the online FFT/iFFT as a significant overhead. Lots of iterative solvers such as Krylov subspace-based method use FFT as a preconditioner for a reason.
> > > > > >
> > > > > > **A** Thank you for this suggestion, we have now reformulated this section. In short, we claimed that GIT-Net outperforms FNO in evaluation cost for large-scale problems by counting their floating-point operations. When using $N$ sampling points of input and output functions, the scaling of an FFT is of the order of $N\log(N)$. Consequently, the evaluation cost of FNO is $\mathcal{O}(N\log (N))$ while GIT-Net's evaluation cost is $\mathcal{O}(N)$. The simulations also demonstrate this in Figure 25.

---

> > > > > > > ### Comment · Reviewer_gw94 · 2023-11-03
> > > > > > > **Sorry for the delay**
> > > > > > >
> > > > > > > Please pardon my delayed responses due to ICLR reviews. I read the revision and I vote to accept the paper.

---

### Review · Reviewer_CRkN · 2023-07-02

**Summary Of Contributions:**

This manuscript proposes a new neural network architecture to learn operators between function spaces in the context of solutions to PDE. The methodology draws inspiration from both PCA-based recent alternatives (which allows them to be robust to the choice of the discretization mesh) as well as the Fourier Neural Operator approach (which seeks to find efficient representations for classes of functions). The authors demonstrate and benchmark their approach, dubbed GIT-Net, in 5 different problems and compare with relevant recent alternatives for data-driven Operator Learning techniques, studying the resulting approximation error and analyzing computational complexities.

**Audience:**

Yes

**Broader Impact Concerns:**

There's no Broader Impact Statement.

**Claims And Evidence:**

Yes

**Requested Changes:**

Main observations

1. As far as I understand, the main novelty of this method is in the introduction of the GIT mapping (eq 8 and 9). While the authors have a descriptive motivation (but only high-level and heuristic) on why the chosen parametrization of \Kappa might make sense, it is unclear why the authors then choose to do away with this by choosing the mapping (T+\Kappa), for a linear operator T. The authors mention that this seems to help numerically from an optimization perspective, but to me this is a bit unsatisfying: If the operator T is so important for the good performance of the obtained trained model, is \Kappa needed at all? Why couldn’t one simple define an operator \tilde{T} = T+\Kappa, and simply train \tilde{T} from data? In other words, what is the added value to the heuristic motivations in page 4 and 5? I would like to see the authors discuss this in detail, and demonstrate that employing T+\Kappa, instead of simply learning a linear operator \tilde{T}, is beneficial.

2. As a continuation of the above, it is a bit unclear how GIT-net differs from other approaches. In particular, the PCA-net approach composes two PCA-based transformations with an MLP (multi layer perceptron). But from the discussion above, there exist an operator \tilde{T} = T + \Kappa, and therefore their GIT-net also composes PCA transformations with an MLP. Therefore, it is unclear whether their improvement over PCA-net is simply because of a different parametrization that allows for -somehow- better optimization? The authors do mention that when C=1, then PCAnet and their method are equivalent, but in light of this -and respectfully- the novelty of the proposed approach seems quite limited - GIT-net simply allows for more channels?

3. When comparing the numerical performance of the different methods, the authors first present the test error of the different approaches (for each of the different problems) as a function of some hyperparamter than, in each case, controls the size of the model. The authors seem want to make an argument for over-fitting (or lack thereof), but without incorporating statistics of the training error, it is impossible to make claims about overfitting. Can the authors add or comment on this? Moreover, the comparison among models is a bit strange, since each x-axis (for each column) is different and it is unclear how to relate each access to the other. Some natural way might be, say, the minimum number of parameters (as a function K, C, or combinations thereof) that make the training error zero.

4. In comparing the error profiles achieved by different methods, the authors chose to present two cases per approach: the one with minimum error and the one with maximum error. I don’t find these comparisons useful, for two reasons: a) this leads to different samples for different methods, and b) it is sometimes unclear how relevant this distinction is, given that often the difference in error between the maximum and minimum cases is very very small/subtle.
This reviewer thinks it would be drastically more useful to i) provide error profiles for different methods for the same sample(s), thus allowing the reader to more easily compare across methods, and ii) provide full statistics for the distributions of those errors; i.e. Fig. 7 only provides the mean, and Figs 8-11 only provide illustration of max and min errors. It would be better to provide a histogram, or at least some illustration with confidence intervals, of the errors provided by each method. This would allow important conclusions, on whether e.g. these distributions are more or less concentrated.

5. Fig. 13, which aims to reflect computational costs, is unclear to me: how is it that test error increases as more computations are used?

Minor comments:

i) authors might want to use more standard notation for the inner product in Eq. 13, such as < . , .>

ii) page 7: “In the numerical presented in..” → numerical experiments presented in?

iii) page 2: “furthermore” → Furthermore.

**Strengths And Weaknesses:**

Strengths

- This paper studies an important and timely problem, on how to approximate solutions to PDE in an efficient and data-driven manner.
- The detailed discussion of related works and approaches is highly appreciated, as it provides a comprehensive background on alternative methods.
- The proposed approach compares favorably, alas mildly, with the alternatives.
- The paper is quite clearly written, and the presentation is generally balanced

Weaknesses

- The modification presented in this manuscript, and that leads to the proposed GIT architecture, seems pretty minor with respect to existing alternatives. In particular, it is not completely clear to this reviewer what the specific changes are (see below).
- While the authors have some heuristic motivations for the construction of their operator \Kappa, little is mentioned about why the function \sigma( T\alpha + \Kappa \alpha), or their multiple compositions, might be doing to the obtain a solution as a function of \alpha.
- In my view, the presentation of the numerical results does not clearly present clear comparisons between the reported methods (see below).

---

> ### Author Response · Authors · 2023-07-30
> **Response to Reviewer CRkN (1)**
>
> [**Main observations**]
>
>   **Q1.** As far as I understand, the main novelty of this method is in the introduction of the GIT mapping (eq 8 and 9) ......
>
>   **A** In our proposed GIT-Net, we build upon the success of FNO, which excels in operator learning due to the incorporation of the kernel integral operator $\mathcal{K}$ in its block $\sigma(W + \mathcal{K})$. FNO leverages the Fourier transform to efficiently solve partial differential equations by transforming them into solvable algebraic equations using Fourier techniques, and subsequently applying the Fourier inverse transform to obtain the original solutions. The linear transformation $W$ acts on the channel direction, while $\sigma$ represents a nonlinear activation function.
>
>   To overcome the limitations of Fourier transform, such as its applicability only to specific grids, in GIT-Net, we generalize FNO by employing a more general integral transform that is parameterized by neural network in $\mathcal{K}$. This alternative integral transform offers a broader range of linear transformations compared to the Fourier transform, as described in Equation (5). The frequency domain product is defined in Equation (6).
>
>   Same with $W$ in FNO, $T$ in GIT-Net also represents a linear transformation in the channel direction. $\mathcal{K}$ in GIT-Net and FNO allows for more complex kernel integral transformations. The combination of $T$ with $\mathcal{K}$ and $\sigma$ in GIT-Net follows the same structure as FNO, and we also explored an alternative structure of $T + \sigma \mathcal{K}$. Comparing the quantization results (see Appendix B, Figure 26 in the updated manuscript), we observed that the former structure, $\sigma(T + \mathcal{K})$, tends to demonstrate a slight advantage, particularly with larger networks and abundant training data. However, overall, the prediction errors between the two structures remain similar.
>
>   By employing this approach, GIT-Net harnesses the strengths of the generalized integral transform, enabling more flexible and powerful transformations, while maintaining a similar architecture to FNO, which has proven effective in operator learning tasks.
>
>   **Q2.** As a continuation of the above, it is a bit unclear how GIT-net differs from other ......
>
>   **A** The novelty of GIT-Net not only lies in multiple channels, and also because its operator $\mathcal{K}$ is an analogy to PDE solvers based on integral transforms and allows a wider search of integral kernel than any fixed integral transforms.
>
>   GIT-Net and PCA-Net both use PCA reduction. However, they differ in their intermediate representations and transformations.
>
>   PCA-Net has a one-dimensional vector intermediate representation with a pointwise linear transformation. In contrast, GIT-Net has a multi-channel two-dimensional matrix representation with dimension $C\times K$ and involves a combination of pointwise and depthwise linear transformations, including a neural network parameterized operator, $\mathcal{K}$, inspired by the Fourier transform for solving partial differential equations.
>
>   PCA-Net and GIT-Net are equivalent when $C=1$, because the depthwise transformation reduces to a scalar product, resulting in $\mathcal{K}$ becoming a pointwise linear transformation, and $T$ also remains a pointwise linear transformation. Consequently, GIT-Net becomes equivalent to PCA-Net. However, for $C>1$, GIT-Net's deeper structure and the neural network parameterized $\mathcal{K}$ result in superior fitting ability compared to PCA-Net. Simply increasing PCA-Net's size to match GIT-Net's intermediate representation does not yield similar predictive performance.
>
>   In summary, GIT-Net's advantage lies in its incorporation of the parameterized operator $\mathcal{K}$, providing enhanced fitting ability, particularly when $C>1$, thanks to its deeper structure inspired by the Fourier transform.

---

> > ### Author Response · Authors · 2023-07-30
> > **Response to Reviewer CRkN (2)**
> >
> > **Q3.** When comparing the numerical performance of the different methods, the authors first present the test error of the different approaches ......
> >
> >   **A** There may indeed be overfitting in the simulation experiment, but it is not the primary focus of our article. Our main objective is to explore neural networks' capability to predict PDE solutions through training. As such, we solely compare their prediction errors without displaying the training errors.
> >
> >   The horizontal axis in our comparison represents the hyperparameters, allowing us to investigate their influence on prediction errors and assess the sensitivity of these errors to model hyperparameters. Despite varying horizontal axes due to different hyperparameters for each neural network, Figure 9 demonstrates that prediction errors change within hyperparameter ranges. For instance, as shown in Figure 9 (updated version), increasing the hyperparameters of PCA-Net and Pod-DeepOnet eventually leads to a plateau or even an increase in prediction error, while FNO and GIT-Net continue to exhibit declining prediction errors with increasing hyperparameters.
> >
> >   **Q4.** In comparing the error profiles achieved by different methods, the authors chose to present two cases per approach: the one with minimum error and the one with maximum error .......
> >
> >   **A** Thank you for your valuable suggestions. We agree that presenting histograms of test errors would be a beneficial addition to our analysis. As per your recommendation, we have now included histograms of test errors, error profiles for the same sample, and specific cases with median and largest test errors in the manuscript. These additions aim to provide a comprehensive view of the test error distribution and offer detailed insights into the errors observed. We appreciate your input, and the manuscript has been updated accordingly.
> >
> >   **Q5.** Fig. 13, which aims to reflect computational costs, is unclear to me: how is it that test error increases as more computations are used?
> >
> >   **A** As computational costs increase with the increment of hyperparameters, larger neural networks often exhibit enhanced expressiveness. However, this can also lead to potential issues such as overfitting or optimization challenges, resulting in an increase in test error. As evident from Figure 9, there are instances where the test error rises with the growth of hyperparameters, indicating an associated increase in computational costs.
> >
> >
> >
> >   [**Minor comments:**]
> >
> >   **i)** authors might want to use more standard notation for the inner product in Eq. 13, such as < . , .>
> >
> >   **A** The double-dot product ":" in Eq. 13 denotes the inner product between two second-order tensors (summation over repeated indices is implied). This inner product notation "<. , .>" usually denotes the inner product between vectors. The double-dot product is a standard notation for the inner product between two second-order tensors in the field of linear elasticity, see following references.
> >
> >   [1] *William S Slaughter. The linearized theory of elasticity. Springer Science & Business Media, 2012.*
> >
> >   [2] https://en.wikipedia.org/wiki/Linear_elasticity
> >
> >   **ii)** page 7: “In the numerical presented in..” → numerical experiments presented in?
> >
> >   **A** It was corrected as you suggested.
> >
> >   **iii)** page 2: “furthermore” → Furthermore.
> >
> >   **A** It was corrected as you suggested.

---

### Public Comment · ~Marimuthu_Kalimuthu1 · 2023-02-27
**Some suggestions for improvement**

**Typos:**
a loss functions => a loss function
In prior work => In prior/previous/earlier works
furthermore, interpolation => Furthermore, interpolation
to efficiently parametrized =>  to efficiently parametrize
consisting in defining => consisting of defining
evaluation in previous work => evaluation in previous works
PCA-NetBhattacharya => PCA-Net Bhattacharya
contrarily to => contrary to
MAD => Meta-Auto-Decoder (MAD)    # for more clarity

In section **our contribution**:

partial differential equations (PDEs) => PDEs   # already shortened two times in the previous sections.
Principal Component Analysis (PCA) => PCA   # already shortened
A comparison of the GIT-Net =>  A comparison of GIT-Net

**Nice to have**:
A pictorial representation of the GIT-Net architecture included, for example in Sec 2.2, would be appreciated for clarity.
Although the plots are added, a tabulated comparison of nRMSE values would be a worthy addition for clarity.
Experiments on the comprehensive PDEBench datasets (https://github.com/pdebench/PDEBench) and, if possible, a comparison with UNet/U-shaped Neural Operator (UNO).

---

### Decision · Action_Editors · 2023-11-03

**Recommendation:** Accept with minor revision

**Comment:**

This work has been positively evaluated by three referees, who highlight the importance of the topic. That said, they all raise the fact that this work is particularly incremental with respect to the literature, so it is particularly important that the authors, in their final revision, make an extra effort to place this work in the literature and clarify in an objective manner the similarities and novelties.

**Audience:**

The topic of this submission is the nascent subfield of 'Operator Learning', which aims to transfer methods that learn in high-dimensional spaces to infinite-dimensional spaces with appropriate non-parametric notions of regularity. As such, the intended audience is the scientific ML.

**Claims And Evidence:**

The paper's claims are indeed backed by sufficient empirical evaluations.